

# Quantifying the effects of noise in a quantum convolutional neural network

Zeyu Fan[1][*], Jonathan Wei Zhong Lau[1][†] and Leong-Chuan Kwek[1,2,3,4]

**1** Centre for Quantum Technologies, National University of Singapore 117543, Singapore
**2** MajuLab, CNRS-UNS-NUS-NTU International Joint Research Unit, UMI 3654, Singapore
**3** National Institute of Education, Nanyang Technological University,
1 Nanyang Walk, Singapore 637616
**4** School of Electrical and Electronic Engineering Block S2.1,
50 Nanyang Avenue, Singapore 639798

[*] e0497425@u.nus.edu , [†] e0032323@u.nus.edu

## Abstract

This study quantifies the effects of quantum noise on the performance of a quantum convolutional neural network (QCNN), building on parallels with classical convolutional neural networks (CNNs), where added Gaussian noise can improve training speed, accuracy, and generalizability. While such benefits are established for classical CNNs, the influence of noise on quantum counterparts remains insufficiently characterized. We specifically examine three types of quantum noise: decoherence, Gaussian noise arising from imperfect quantum gates and experimental error, and systematic noise introduced during input state preparation. Our analysis provides a detailed assessment of how these distinct noise sources affect QCNN operation and outlines considerations for mitigating their impact. Though a QCNN is used as an example in this work, the methods used here can be applied to other quantum machine learning models as well.

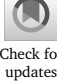

# 1 Introduction

Neural networks are a machine learning technique inspired by the architecture of neurons in biological systems. Yet, they are a far cry from their original biological counterparts. Today, machine learning (ML) has found many widespread applications such as gravitational wave astronomy [1–3], medical imaging [4, 5], data classification [6–8], and experimental particle data [9]. Recently, machine learning techniques have also been widely applied in quantum technologies [10–12]. Moreover, many examples of such algorithms have been successfully implemented on quantum computers using quantum input data [13].

Indeed, the convolutional neural network is historically one of the first success stories of deep machine learning. An eye-catching success of convolutional neural network architectures stems from their ability to tackle image-classification tasks some ten years ago in long-standing benchmarks like ImageNet [6]. Convolutional neural networks employ a technique known as convolution—essentially a dot-product operation between a grid-structured set of weights and similar grid-structured inputs drawn from different spatial localities in the input volume. Such an operation is often useful for data with a high spatial or other locality level, such as image data.

Even with GPUs, machine learning techniques in general require huge computing power. Since quantum computing promises to provide a solution to the increasing demand for computational speed and capacity, it is not surprising that many researchers have studied the possibility of extending machine learning to the quantum regime [14–16]. But one of the greatest challenges that near-term quantum computers face is quantum noise. Current physical quantum qubits are inherently unstable, and any interactions with the environment can perturb them and lead to decoherence. Optical qubits, while mostly safe from interactions with the environment, may similarly be affected by optical elements and detectors that are not yet of perfect performance.

In classical machine learning, however, the introduction of Gaussian noise has been shown to provide certain advantages. Gaussian noise in the input data is used to prevent overfitting [17, 18], can improve training speed [19], and may help the gradient descent process escape shallow local minima into deeper minima, improving model performance [20]. Whether similar effects arise in quantum machine learning models, specifically in quantum convolutional neural networks, remains to be determined.

In this work, we focus on quantifying the effects of noise on quantum convolutional neural networks, rather than investigating whether such noise offers any inherent performance advantages. We examine the impact of three different types of noise separately: (i) quantum noise and decoherence modeled by the quantum depolarizing channel, (ii) Gaussian noise arising from imperfect gates in a quantum circuit and measurements in experimental implementations (shot noise), and (iii) systematic quantum noise in the form of perturbations to input states during state creation.

Our aim is to provide a clear assessment of how each of these noise sources affects the operation of quantum convolutional neural networks. We also discuss considerations for mitigating these effects, particularly in experimental implementations of such quantum circuits.

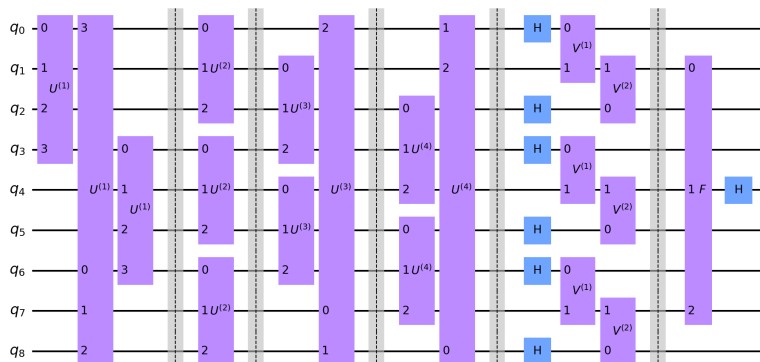

Figure 1: Example graphic representation of the QCNN architecture, with $N_p = 3$ and 9 total qubits. Qubits are evolved through the circuit from left to right, though unitary operators represented by purple blocks. The numbers aligned along the left of each block represent the qubits on which they act. The blue (H) blocks are Hadamard gates, which transform the qubits into the $x$-basis. The expectation value of the qubit $q_4$ at the end of the circuit is taken as the output of the QCNN in this case.

## 1.1   Quantum convolutional neural network (QCNN)

In a system of qubits, a quantum state $|\psi\rangle$ is some linear combination of its possible basis states, which can be represented in statevector form:

$$|\psi\rangle = a_1 |...000\rangle + a_2 |...001\rangle + a_3 |...010\rangle + a_4 |...011\rangle + ... \equiv \begin{pmatrix} a_1 \\ a_2 \\ a_3 \\ a_4 \\ \vdots \end{pmatrix}. \qquad (1)$$

We can utilize the amplitudes of each state, $a_i$, as a quantum system analogous to neurons in classical machine learning. Similarly, unitarily transforming such quantum states can be akin to matrix transformations in classical neural networks.

Each neuron in a classical CNN samples a subset of the input through a convolution kernel—the QCNN will accomplish this through unitary transformations across several neighboring qubits, repeated to encompass all qubits in a single layer. In our work, we use the QCNN architecture in [21] and implemented in Qiskit [22].

The structure of this QCNN is determined by the total number of qubits, as well as the number of qubits to be pooled in the pooling layer $N_p$. The first convolution layer applies a unitary over $N_p + 1$ qubits on every $N_p^{th}$ qubit, followed by $N_p$ convolution layers of unitaries across $N_p$ qubits in the same manner. The pooling layer measures $N_p - 1$ qubits in the $X$-basis, which then applies a unitary operator on the remaining qubit depending on the outcome of the measurement. All remaining qubits go through the final fully connected layer, and the hypothesis (or output) from the QCNN is taken as the expectation value of one of these remaining qubits, as measured in the $x$-basis.

A graphic example of the QCNN shown in Fig. (1), with $N_p = 3$ and 9 qubits, and is used for our quantum dataset.

The unitary operators are parameterized through the coefficients of the generators of the $SU(n)$ group for a $n$-qubit operator, so each layer contains $2^{2n} - 1$ trainable parameters. A unitary $U^{(l)}$ on layer $l$ (corresponding to $U_l$ on Figure (1)) can be obtained from these generators $\Lambda_i^{(l)}$, known as the generalized Gell-Mann matrices, and their coefficients ($c_i^{(l)}$) through

the exponential map:

$$U^{(l)} = \exp\left(\sum_{i=1}^{2^{2n}-1} -c_i^{(l)} \Lambda_i^{(l)}\right).\tag{2}$$

As such, $c_i^{(l)}$ are the trainable parameters of our QCNN. Similar to gradient descent for weights as in classical NNs, the training process for this QCNN will aim to find the set of $c_i^{(l)}$, $\mathbf{c}$, that minimizes some loss function $L$ over a set of input statevectors $|\psi\rangle_{\text{in}}^{(1,2,\dots N)}$ and their corresponding target labels $y^{(1,2,\dots,N)}$ of the dataset.

$$\mathbf{c} \equiv \underset{\mathbf{c}}{\arg\min}\, L(h(|\psi\rangle_{\text{in}}^{(1,2,\dots N)}; \mathbf{c}), y^{(1,2,\dots,N)}).\tag{3}$$

The hypothesis (or output) from the QCNN from an input statevector $|\psi\rangle_{\text{in}}$, given a set of parameters $\mathbf{c}$ can thus be represented as:

$$h(|\psi\rangle_{\text{in}}; \mathbf{c}) \equiv \langle\psi|_{\text{in}} T^{\dagger}(\mathbf{c}) X_m T(\mathbf{c}) |\psi\rangle_{\text{in}}.\tag{4}$$

Here, $T(\mathbf{c})$ would represent some transformation upon the input state $|\psi\rangle$ as prescribed by the whole QCNN, and $X_m$ is the measurement operator for the measured qubit $m$ in the $x$-basis. We can associate the measurement of the $|+\rangle$ state with the numerical value $+1$, and the measurement of the $|-\rangle$ state with the value -1. Over many repetitions of the circuit, the output of the QCNN is taken as the expectation value of these measurements, which will be bounded within [-1,1]. Should the final state of the measured qubit be in a state $|\psi_{\text{result}}\rangle = \alpha\,|+\rangle + \beta\,|-\rangle$ for some $(\alpha, \beta)$, the measured output would have an expectation value of:

$$h(|\psi\rangle_{\text{in}}; \mathbf{c}) = |\alpha|^2 - |\beta|^2 \equiv 2|\alpha|^2 - 1.\tag{5}$$

This will be the output for the Quantum Dataset. The Classical dataset instead uses ML tasks that have target values outside of this range can be normalized to values in [-1,1]. In this QCNN, non-linearities are introduced in the measurement processes, in the pooling layer, and in the final measurement of the qubit. These measurements do not take the amplitudes in a state $|\psi\rangle$ directly, but rather the squared absolute value of amplitudes of several basis states in the statevector. The sum of the squared absolute values would be a non-linear function of the original amplitudes, which allows the QCNN to model non-linear relationships much alike activation functions in classical neural networks.

The details of this QCNN and our implementation are elaborated in section 9.

## 2 Datasets

In this section, we will introduce the datasets we will be using in our testing. In addition to a quantum dataset that uses quantum states as inputs, we will also explore the effect of noise on a QCNN that handles classical data encoded into quantum states.

### 2.1 Quantum dataset: Quantum phase recognition

We use the quantum phase recognition task from [21], which the QCNN was originally designed for, as the quantum dataset. The QCNN is tasked to recognize the symmetry-protected topological (SPT) phases in a 1-dimensional spin chain governed by the Hamiltonian:

$$H = -J \sum_{i=1}^{N-2} Z_i X_{i+1} Z_{i+2} - h_1 \sum_{i=1}^{N} X_i - h_2 \sum_{i=1}^{N-1} X_i X_{i+1},\tag{6}$$

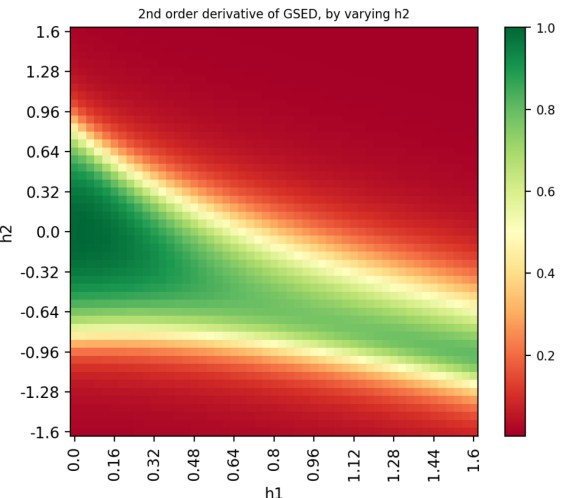

Figure 2: Phase diagram of ground state wavefunctions from the Hamiltonian in eq. (6), with $J = 1$. The colour indicates the second order partial derivative of ground state energy density along $h_2$, different from the binary output in Ref. [21]. This also identifies the phase transition area and provides a good estimate of the SPT phase in Figure 2(a) of Ref. [21].

with $X_i$ and $Z_i$ being the $x, z$ pauli operators for the spin at the $i^{th}$ site. For our Hamiltonian in eq. (6), a phase transition occurs in the ground state as $h_1/J$ and $h_2/J$ changes, at which exists a $\mathbb{Z}_2 \times \mathbb{Z}_2$ symmetry-protected topological phase.

The ground state statevectors at various values of $h_1/J \in [0, 1.6], h_2/J \in [-1.6, 1.6]$ are passed as the input data to the QCNN. We use 9 qubits to simulate the spin chain of length 9, as per the original work [21].

We can also identify this SPT phase by the second order derivative of the ground state energy density (GSED) along $h_2/J$, which indicates the area where the phase transition occurs. Each input ground state statevector would have a corresponding target label in $[0, 1]$. Fig. 2 shows the target labels which the QCNN is tasked to identify, indicating this GSED for the ground states at each value of $h_1/J$ and $h_2/J$.

As such, the output of the QCNN will be the predicted second order partial derivative of GSED, and this output can be obtained from the QCNN via measurement of one of the qubits as per eq. (4).

For this task, a mean squared error loss is used. This is defined as:

$$L(h[|\psi\rangle_{\text{in}}^{(1,2,...N)}; \mathbf{c}], y^{(1,2,...,N)}) = \frac{1}{N} \sum_{i=1}^{N} \left\{ h[|\psi\rangle_{\text{in}}^{(i)}; \mathbf{c}] - y^{(i)} \right\}^2 . \tag{7}$$

More information on specific implementation decisions in this work for this dataset can be found in section 10.

## 2.2 Classical dataset: MNIST

Since we expect the introduction of noise to possibly help the QCNN be robust to the random noise and errors inherent in real-world datasets, we also test the QCNN on a classical dataset that contains such real-world uncertainties.

We also use the MNIST dataset [23] for this purpose. The MNIST dataset consists of scanned images of handwritten number digits, with each image labelled as the digit it represents.

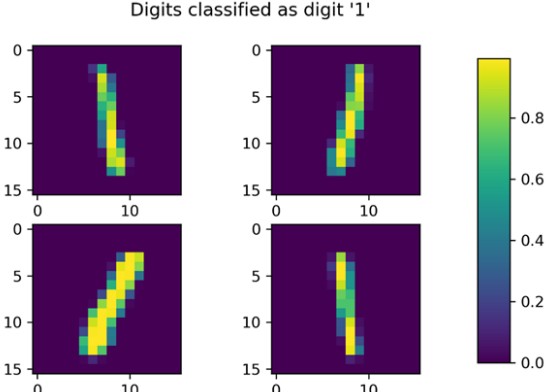

Figure 3: Several example images that are classified as the digit '1' from the MNIST dataset [23], rescaled to $16 \times 16$ pixels from the original $28 \times 28$ pixels. The darkness of each pixel is encoded as a numeric value $\in [0, 1]$ in a $16 \times 16$ matrix—a numeric value of 0 indicates a completely white pixel, and 1 indicates a completely dark pixel.

This classical dataset is encoded into quantum data through amplitude encoding. We flatten the 2-dimensional matrix along its rows into a 1-dimensional vector—the $(i, j)$ element of the original matrix would correspond to the $(N \times i - 1) + j$ component of this vector. These components can be encoded into the amplitudes of the possible quantum states of the input state vector $|\psi\rangle_{\text{in}}$.

$$|\psi\rangle_{\text{in}} = A \sum_{i=1}^{N} x_i |i\rangle \,. \tag{8}$$

Here, $x_i$ is the $i^{th}$ component of the vector representing the classical data, and $|i\rangle$ is the $i^{th}$ computational basis state of the system of qubits. $A$ is the normalization factor, as given by $\frac{1}{\sqrt{\sum_{i=1}^{N} x_i^2}}$. A vector with $2^n$ components can be encoded in the amplitudes of the states spanned by $n$ qubits—an example vector with 4 components $(x_1, x_2, x_3, x_4)$ can be encoded into $|\psi\rangle_{\text{in}} = A(x_1 |00\rangle + x_2 |01\rangle + x_3 |10\rangle + x_4 |11\rangle)$. The target labels for each input statevector would be a binary 0 or 1, $y^{(1,2,\dots,N)} \in \{0, 1\}$.

For this dataset, we opt to use the binary cross-entropy loss [24], or logloss, which is commonly used in classification tasks where the target values take on values only two possible values, such as zeroes or ones, $y^{(i)} = \{0, 1\}$.

$$L(h[|\psi\rangle_{\text{in}}^{(1,2,\dots N)}; \mathbf{c}], y^{(1,2,\dots,N)}) = -\frac{1}{N} \sum_{k=1}^{N} y^{(k)} \log_2 \left( h[|\psi\rangle_{\text{in}}^{(k)}; \mathbf{c}] \right) + (1 - y^{(k)}) \log_2 \left( 1 - h[|\psi\rangle_{\text{in}}^{(k)}; \mathbf{c}] \right). \tag{9}$$

For this task, the output from the QCNN will be defined as $h[|\psi\rangle_{\text{in}}^{(i)}; \mathbf{c}] = |\alpha|^2$ instead of $2|\alpha|^2 - 1$. $h[|\psi\rangle_{\text{in}}^{(i)}; \mathbf{c}]$ in this case will be the likelihood of the input data to represent the digit '1'. The reasoning for the change in retrival of output from the QCNN is given under "Loss Function" in section 11. Section 11 also includes more information on the implementation of this dataset in general.

## 3 Sources of noise

In this section, we shall introduce some sources of noise that may be present in future experimental implementations of the QCNN—such as measurement errors, decoherence of the state as it goes through the circuit, and errors during the creation of input states.

## 3.1 Gaussian noise

One source of error that a quantum circuit may be affected by is measurement errors. Even discounting the intrinsic uncertainty in the measurement of the state due to the probabilistic nature of quantum measurements, known as shot noise, quantum devices are still affected by experimental sources of error. We can possibly model this effect as a Gaussian distribution applied to the supposed probability of measuring the $|+\rangle$ state when measuring a single-qubit state $|\psi\rangle = \alpha|+\rangle + \beta|-\rangle$ when retrieving the output from the QCNN.

$$|\langle +|\psi\rangle|^2 \equiv |\alpha|^2 \rightarrow \mathcal{N}(|\alpha|^2, \sigma^2). \tag{10}$$

This would affect the output of the QCNN as defined in eq. (5), as specified below.

$$h(|\psi\rangle_{\text{in}}; \mathbf{c}) = 2|\alpha|^2 - 1 \rightarrow 2\mathcal{N}(|\alpha|^2, \sigma^2) - 1. \tag{11}$$

As mentioned in the introduction, Gaussian noise introduced to CNNs on classical computers has been found to improve its training speed [19]. In addition, Gaussian noise can potentially prevent overfitting [17] [18]. We shall investigate how various magnitudes of this noise can affect the training and operation of the QCNN.

## 3.2 Quantum noise

One commonly used model for quantum noise is the quantum depolarising channel (DPC), which models the decoherence that a quantum state is subjected to from interactions with the surrounding environment [25]. The depolarising channel evolves the density matrix $\rho$ of some $N$-qubit state as follows:

$$\rho \rightarrow p\frac{I}{2^N} + (1-p)\rho. \tag{12}$$

Physically, it can be interpreted as some process that introduces bit flip, phase flip, and a combination of both errors each with equal probability $p/4$—as such, it also models random, non-systematic gate errors without any bias towards any one specific type of error, when a qubit is measured over many repetitions.

To understand how the DPC can affect the output from the QCNN, the output from the QCNN is found as the expectation value of a measurement $X_m$ on the single qubit $m$, which can be found from the reduced density matrix over this qubit $\rho^m$.

$$h(|\psi\rangle; \mathbf{c}) \equiv \langle X_m \rangle = Tr(\rho^m X_m). \tag{13}$$

We have referenced the notation from eq. (4), with $\rho^m$ being the reduced density matrix after a partial trace over all other qubits of the full density matrix $\rho$ at end of the QCNN, $\rho = T(\mathbf{c})|\psi\rangle\langle\psi|T^\dagger(\mathbf{c})$. After the DPC, $\rho^m$ transforms as per eq. (12), and thus producing the expectation value:

$$\langle X_m \rangle = Tr\left(\left(p\frac{I}{2} + (1-p)\rho^m\right)X_m\right). \tag{14}$$

Since the measurement of $I/2$ about any basis has an expectation value of zero, we see that the output of the QCNN is scaled by $(1-p)$. We shall test and attempt to quantify the effects of this noise on the operation of the QCNN.

## 3.3 Adversarial training

It is also possible that errors may be introduced during the creation of an input state. To this end, we also explore a quantum adversarial training regime as proposed by [26], which

prescribes a perturbation operator $U_\delta$ within some small region $\Delta$ on the input state $|\psi\rangle_{\text{in}}$ to maximize the loss function of the NN.

$$U_\delta \equiv \underset{U_\delta \in \Delta}{\arg\max}\, L(h(U_\delta\,|\psi\rangle_{\text{in}}^{(1,2,\dots N)}; \mathbf{c}), y^{(1,2,\dots,N)})\,. \tag{15}$$

Gradient ascent can be used to find such a perturbation operator that maximizes the loss function. In [26], the perturbation is modeled by its effect on the input statevector, and the amplitude of each computational basis state is changed by a small amount $x_i \rightarrow x_i + \delta_i$ for the $i^{th}$ computational basis state.

We perturb an input statevector by applying single-qubit unitary transformations on each of the input qubits, also parameterized by the coefficients of their generalized Gell-Mann matrices—the unitary operator on the $i^{th}$ qubit would be:

$$U_\delta^{(i)} = \exp\left(-i(\delta_1^{(i)}X + \delta_2^{(i)}Y + \delta_3^{(i)}Z)\right)\,. \tag{16}$$

We can specify the magnitude of the perturbation by limiting the mean of these coefficients to some value $\bar{\delta}$. This is defined as:

$$\bar{\delta} = \frac{1}{3N}\sum_{i=1}^{N}\sum_{j=1}^{3}|\delta_j^{(i)}|\,. \tag{17}$$

The perturbation can be limited by demanding that $\bar{\delta}$ is less than some given value, and we shall test the effect of various magnitudes of perturbation on the QCNN by varying $\bar{\delta}$.

The main benefit of parameterizing the perturbations in this manner would be computational complexity—during gradient descent to find the perturbative parameters that maximize the loss function, instead of finding the derivative for each of the $2^N$ amplitudes of the input statevector for $N$ qubits, this would need to be performed for only $3N$ coefficients. A single-qubit transformation would also be easier for experimental/practical implementation of the quantum circuit.

These perturbations can model "worst-case" systematic errors that an experimental setup can introduce to the input states during creation and initialization, before it is fed through the QCNN.

# 4 Results & discussions

We present the results from applying various sources and magnitudes of noise during the gradient descent/training process. The noise is only applied in calculation of gradients in this process; all presented results in this section use the true value of loss, without any noise applied whatsoever, to illustrate the actual performance of the QCNN and be able to compare the impact of various magnitudes of noise on the training process.

# 5 Quantum dataset

In the absence of noise, we can verify that gradient descent in the QCNN functions as expected, and the trained model is able to consistently identify the SPT phase well, as seen in Fig. (4, 5) as follows.

We shall compare these results in this noiseless case with the results when the training process is subjected to the different sources of noise in the following subsections.

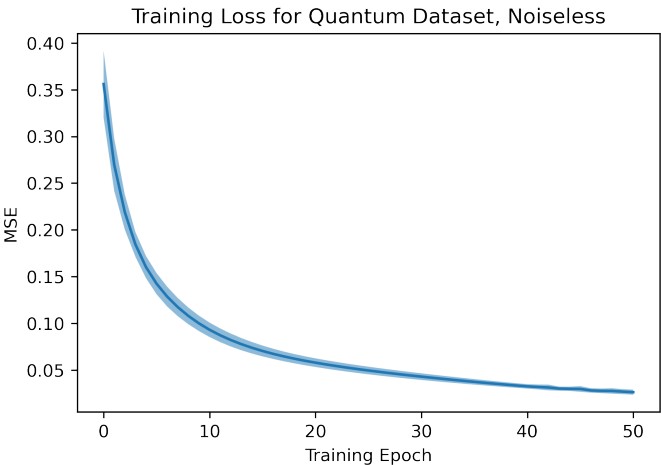

Figure 4: The training-loss plotted over 50 epochs of training, repeated over 10 different sets of initial parameters. The mean squared error (MSE) over the training set is calculated at the end of every training epoch, after parameters are updated. The solid line indicates the mean loss over the 10 runs, and the shaded region indicates the standard deviation. The model is able to attain a very low MSE loss of $0.02626 \pm 0.00090$

Table 1: The mean loss (true MSE, without Gaussian noise) on the training and test datasets for the trained model after 50 training epochs, with Gaussian noise at various values of $\sigma$ introduced during training.

| $\sigma$ | Loss (Training set) | Loss (Test set) |
|---|---|---|
| 0 | 0.02626(90) | 0.0199(13) |
| 1e-3 | 0.351(11) | 0.1933(58) |
| 1e-4 | 0.2054(52) | 0.1201(37) |
| 1e-4.5 | 0.0792(36) | 0.0484(23) |
| 1e-5 | 0.0422(19) | 0.0288(12) |

## 5.1 Gaussian noise

Due to the stochastic nature of the Gaussian noise that is applied, we repeat the training process for 10 different sets of initial parameters generated from random seeds of $0-9$.

The results of the training process are in Fig. (6), and the loss on the training and test sets along with their standard error over the 10 runs are tabulated in table (1). Note that the standard deviation of loss is used in the Fig. to highlight the range of values of loss at each epoch, while standard error is used in the calculation of the loss in the table since it is a final result. The standard error in this case would be $1/\sqrt{10}$ the value of the standard deviation.

Even though noise has been found to be beneficial in classical machine learning, the losses are higher with Gaussian noise applied in all cases, and we were unable to find any benefit from applying Gaussian noise. Even at the small value (compared to usual experimental error) of $\sigma = 10^{-3}$, the Gaussian noise drastically affects the gradient descent process, preventing the loss from decreasing.

At the two values of $\sigma = 10^{-3}, 10^{-4}$, the loss curves flatten out most likely due to the action of the bold-driver mechanism (elaborated at the end of section 9) on the learning rate. Even though the threshold to halve the learning rate is relaxed as described in section (12.1), the fluctuations in loss are still above this threshold for these two cases. Over many epochs, this

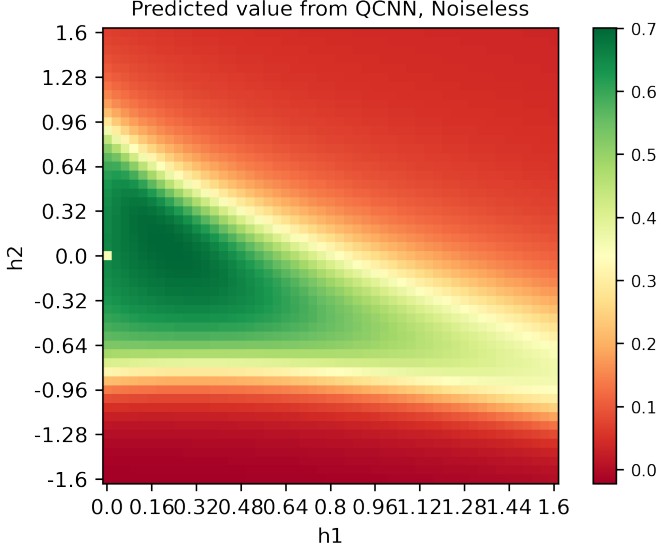

Figure 5: The output/predicted values of the QCNN for one example trained QCNN, when given the wavefunctions at each of the $51 \times 51$ discrete points of the test set as input—the predicted values can be verified to match closely with the target labels similarly plotted in Fig. 2.

causes the learning rate, and consequently any changes in the parameters, to be miniscule. It is only by this mechanism that the losses stabilise at some value—without this, the Gaussian error causes the losses to be increasingly divergent with training epochs.

Though we were unable to find benefits in applying Gaussian noise in this case, we also quantify the effect of such random errors on the operation of the QCNN, to find methods in mitigating its effects.

In Appendix (A), we show the Gaussian noise propagates through the gradient process to cause the parameter updates at each epoch to become a Gaussian variable as well, with the variance of the parameters updates roughly proportional to the below expression:

$$\sigma^2_{\left(\Delta c_i^{(l)}\right)} \propto \left(\frac{2\sigma\eta}{\epsilon}\right)^2 \left[\frac{4\sigma^2 + 8}{N}\right].\tag{18}$$

Here, with $\sigma^2$ being the variance of the Gaussian noise, $\eta$ the learning rate, $\epsilon$ the finite difference parameter when estimating the gradients during gradient descent, and $N$ the number of training examples. For those unfamiliar with the learning rate and the finite difference parameter, a quick explanation is found under "Training Process" in section (9).

One method to reduce this variance is to increase the value of the finite difference parameter $\epsilon$. We illustrate this in Fig. (7).

In the gradient descent process for a parameter $c_i^{(l)}$, we shall define $h^{+(k)}$ as the output value for the $k^{th}$ input when the parameter $c_i^{(l)}$ is increased by a small finite value $\epsilon$, and $h^{-(k)}$ as the output value when the same parameter is decreased by $\epsilon$,

Qualitatively, increasing $\epsilon$ amounts to setting the values of $h^{+(k)}, h^{-(k)}$ to be proportionately further apart, thus reducing the relative effect of random fluctuations in these two values due to the Gaussian noise when taking their difference in the expression for estimating the gradient in eq. (27). We can see that the effect of the Gaussian noise that once caused divergent behaviors in the original $\epsilon = 10^{-4}$ case has been greatly reduced even at $\epsilon = 10^{-2}$, where the training-loss curve approaches that of the noiseless case.

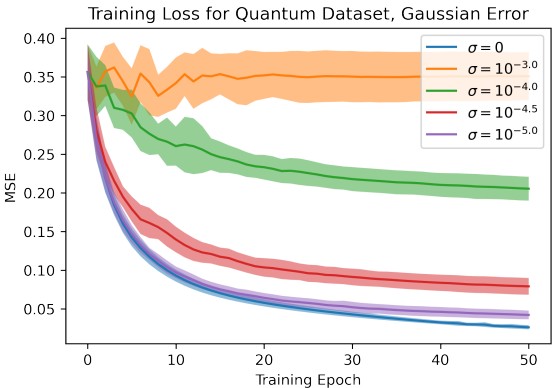

Figure 6: The loss (true MSE over the training set, without any Gaussian noise applied), calculated at the end of every training epoch, with Gaussian noise applied to the measured outputs during the gradient descent process with standard deviation $\sigma$. The solid lines indicates the mean of the loss at each training epoch over the 10 repeated simulations, while the shaded regions indicate the standard deviation.

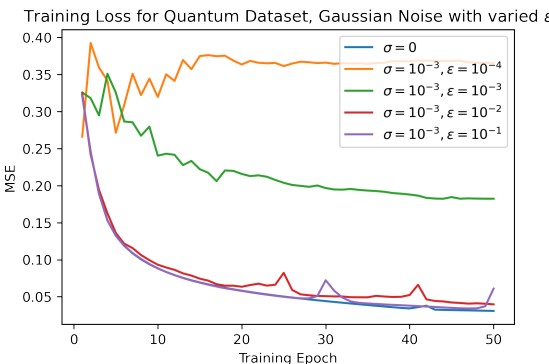

Figure 7: The loss-training plot for a constant Gaussian noise of $\sigma = 10^{-3}$ applied onto the output at various values of $\epsilon$. These are compared to the noiseless case (blue line, $\sigma = 0$).

However, increasing $\epsilon$ will reduce the accuracy in estimating the gradient. This is especially evident near extremal points like minimas, where the second-order derivative gets increasingly significant in the finite difference method when $\epsilon$ is increased. This might be evident in the $\epsilon = 10^{-2}$ and $\epsilon = 10^{-1}$ curves, where we observe more cases of increases in loss that are also greater in magnitude compared to the noiseless case.

## 5.2 Quantum depolarising channel

Fig. (8) shows how loss decreases throughout the training process with DPC applied with different magnitudes of $p$, corresponding to $p_{\text{final}}$ in eq. (33); a measure of the effects of decoherence throughout the entire QCNN circuit on the input states. The final loss on the training and test sets after 50 epochs of training are tabulated in table (2).

While we see only a slight decrease in loss on the test set in the $p = 0.05$ compared to the noiseless case, this is not true on average. If we repeat the training process over 10 different sets of initial parameters using random seeds 0-9 as in the Gaussian noise case, we find that there is no decrease in either on average, this is shown in appendix (B). We are also unable to find any benefit from the noise in this case.

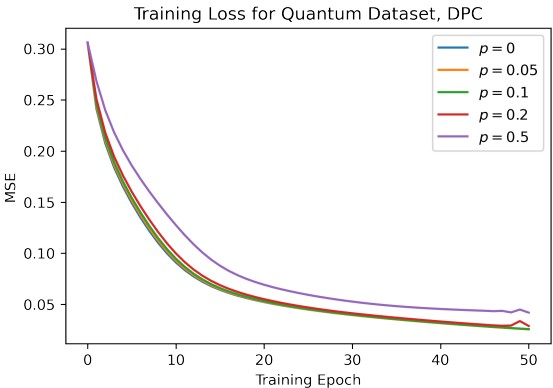

Figure 8: The loss (true mean-squared error over the training set, without DPC applied), calculated at the end of every training epoch, with DPC of various values of the probability $p$ applied onto the output during the gradient descent process.

Table 2: The loss (true MSE, without DPC applied) on the training and test datasets for the trained model after 50 training epochs.

| $p$ | Loss (Training set) | Loss (Test set) |
|---|---|---|
| 0 | 0.02568 | 0.02303 |
| 0.05 | 0.02553 | 0.01862 |
| 0.1 | 0.02580 | 0.02320 |
| 0.2 | 0.02886 | 0.02660 |
| 0.5 | 0.04191 | 0.03809 |

However, it is clear that at low values of $p$, the training process remains mostly unaffected—the $p = 0.05$ line almost perfectly overlaps the $p = 0$ line. The gradient descent process proves to be robust against decoherence and gate errors, as long as such errors are averaged over many repeated runs which the DPC represents. Considering that there are 6 layers in total in our QCNN in this case (4 convolution + 1 pooling + 1 fully connected), the maximum tested value of $p = 0.5$ would be akin to having constant decoherence of $p_i = 0.10910$ at each layer, as explained in section (12.2).

To explain these observed differences, we will attempt to quantify the effect of this noise on the training process. We observe that as mentioned in (3.2), a DPC of magnitude $p$ results in a uniform scaling of all predicted values by $(1 - p)$:

$$h[|\psi\rangle_{in}; \mathbf{c}] \rightarrow (1 - p)h[|\psi\rangle_{in}; \mathbf{c}]. \tag{19}$$

In Appendix (C), we derive the ratio between the gradient with DPC, $\left(\frac{\partial L}{\partial c_i^{(l)}}\right)_p$, and what the gradient would be without DPC $\left(\frac{\partial L}{\partial c_i^{(l)}}\right)_{p=0}$, when all parameters are kept the same between the two cases:

$$\frac{\left(\frac{\partial L}{\partial c_i^{(l)}}\right)_p}{\left(\frac{\partial L}{\partial c_i^{(l)}}\right)_{p=0}} = 1 - p \left[1 + \frac{(1-p)\sum_{k=1}^{N}\left\{\left(h^{+(k)} - h^{-(k)}\right)\left(h^{+(k)} + h^{-(k)}\right)\right\}}{\sum_{k=1}^{N}\left\{\left(h^{+(k)} - h^{-(k)}\right)\left(h^{+(k)} + h^{-(k)} - 2y^{(k)}\right)\right\}}\right]. \tag{20}$$

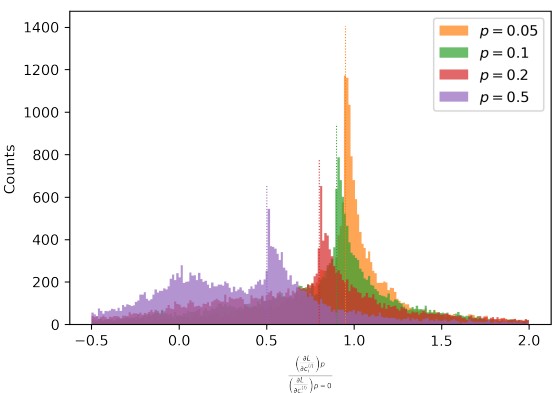

Figure 9: Histograms of the value corresponding to the expression shown in eq. (20), for every calculated gradient in the 50 training epochs for each value of $p$ shown in Fig. (8). We highlight the value $(1-p)$ along the $x$-axis for each histogram using a dotted vertical line of the same colour. The histogram is limited to values in the range $[-0.5, 2]$, which is able to include more than 80% of all 25650 values in all 4 cases.

Once again, in the gradient descent process for a parameter $c_i^{(l)}$, we define $h^{+(k)}$ as the output value for the $k^{th}$ input when the parameter $c_i^{(l)}$ is increased by a small finite value $\epsilon$, and $h^{-(k)}$ as the output value when the same parameter is decreased by $\epsilon$. This is explained under "Training Process" in section (9).

Note that this expression does not represent the cumulative effects of applying DPC on the gradient descent process over multiple epochs, but rather the "instantaneous" change caused by the DPC at each epoch, when compared to the noiseless value without DPC.

The coefficient of $p$ in the square brackets of the expression above are distributed about a value slightly below unity for all parameters, by inspection. To illustrate this point, we plot a histogram of this term for every gradient that is calculated throughout all 50 training epochs for each of the cases shown in Fig. (8) with DPC applied. Since the coefficient of $p$ is slightly below one, the ratio $\left( \frac{\partial L}{\partial c_i^{(l)}} \right)_p / \left( \frac{\partial L}{\partial c_i^{(l)}} \right)_{p=0}$ is distributed about some value slightly above $(1-p)$.

As such, most gradients by which the parameters are updated will get scaled by approximately $(1-p)$ from their original gradient. This explains how the loss decreases at a slower rate as $p$ is increased, as parameters would tend to shift less as $p$ is increased, compared to the noiseless case. Cumulatively, this would lead to the training-loss of higher values of $p$ lagging behind the cases with lower $p$, as we observed in Fig. (8).

This might lead one to the conclusion that it will take proportionately longer for the model to be trained if $p$ is increased—that a process with $p = 0.5$ may take twice as long for loss to decrease below a certain value compared to the $p = 0$ case, since parameters as shifted half as much in the former case. In Fig. (8), the $p = 0$ case reaches below $MSE < 0.10$ by the ninth epoch, while the $p = 0.5$ case reaches this threshold by the 14th epoch. This is because the loss is often not a linear function of the parameters—if shifting a parameter by $x$ produces a shift in loss of $\Delta L$, a shift by $x/2$ may not necessarily produce a shift by $\Delta L/2$, for example.

One last interesting feature in Fig. (8) is the slight uptick in the training loss towards the end of the training process for the $p = 0.2, 0.5$ cases. To explain this, we can explore the difference in how the parameters are updated when DPC is applied. At low values of loss, which is the case after many epochs of training, the predicted values approach the target values—$h^{+(k)}, h^{-(k)} \approx y^{(k)}$. In the noiseless case, the parameter update terms within the square brackets of eq. (26) would approach zero, $\left(h^{+(k)} - y^{(k)}\right) \approx 0$, and $\left(h^{-(k)} - y^{(k)}\right) \approx 0$, resulting

Table 3: The loss (true MSE, without the adversarial perturbations applied) on the training and test datasets for the trained model after 50 training epochs, with the perturbation applied over the inputs while bounded by $\bar{\delta}$.

| $\bar{\delta}$ | Loss (Training set) | Loss (Test set) |
|---|---|---|
| 0 | 0.02799 | 0.02403 |
| 0.01 | 0.02968 | 0.02525 |
| 0.05 | 0.04409 | 0.03877 |
| 0.1 | 0.08480 | 0.06415 |
| 0.5 | 0.30128 | 0.16499 |

in near-zero parameter updates.

In the case with DPC, the parameter update term in the square brackets of eq. (C.2) differs due to the presence of the factor $(1-p)$ on $h^{+(k)}$ and $h^{-(k)}$. At higher values of $p$, such as the $p = 0.2$ and $p = 0.5$ case, the parameter update term remains significant even when $h^{(k)} \approx y^{(k)}$. Consequently, large values of $p$ can lead to parameter overshooting beyond the minima, a behavior not observed without DPC.

In addition, when this occurs, the coefficient of $p$ in eq. (20) is far from unity—the denominator in the fraction of this term would approach zero as $\left(h^{+(k)} + h^{-(k)} - 2y^{(k)}\right) \to 0$ when $h^{(k)} \approx y^{(k)}$, while the numerator may remain finite. This phenomenon also explains why the histograms in Fig. (9) are bounded within the range $[-0.5, 2]$, as extreme values exceeding 1000 exist.

Though the noise does not affect the training process much, should one want to mitigate this source of error, one approach can be adjusting the predicted value to account for the change caused by the DPC. If an experimental setup is found to have decoherence at some specific value of $p$, the predicted value can be multiplied by $1/(1-p)$.

## 5.3 Adversarial training

We limit the perturbation by the adversary via bounding the 1-norm of the perturbative parameters by their absolute mean $\bar{\delta}$, $\frac{1}{3N}\left(\sum_{i=1}^{N}\sum_{j=1}^{3}|\delta_j^{(i)}|\right) \leq \bar{\delta}$, as described in section 3.3. For a QCNN with 9 qubits, there will be a total of 27 of such parameters, which are then trained by gradient descent as described in section (12.3).

The application of this perturbation is also fully deterministic in the training process, given the set of initial parameters; a set of initial parameters $c_i^{(l)}$ together with perturbative parameters $\delta_j^{(i)}$ will always end up in the same minima and maxima respectively, at the end of the training process. Though, different sets of initial parameters will end up in different minima/maxima—to ensure consistency once again, we intialise all $c_i^{(l)}$ and $\delta_j^{(i)}$ using a random seed of 1.

Fig. (10) shows how loss decreases throughout the training process when the perturbation is limited to various values of $\bar{\delta}$, and the results from the trained models are in table (3).

We found no consistent benefit in terms of final model performance. While a slightly faster decrease in loss was observed in the first few epochs at small values of $\bar{\delta}$ ($\bar{\delta} = 0.01, 0.05$ in the inset of Fig. (10)), this effect is offset by the additional computational cost required to optimize the perturbative parameters.

Our results agree with the conclusions of [26], [27]—that the QCNN is also found to be vulnerable to even small values of adversarial perturbations.

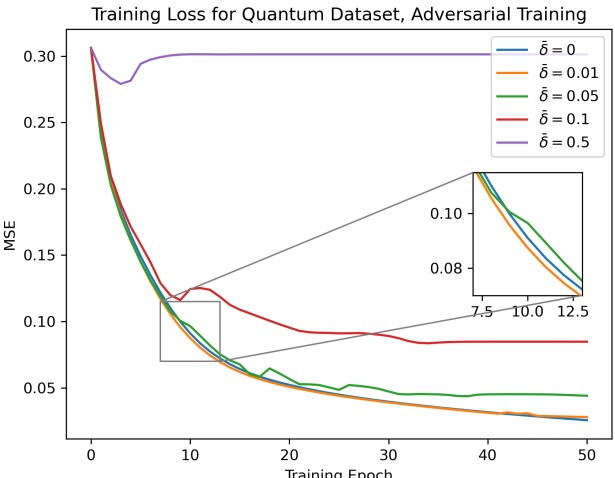

Figure 10: The loss (true mean-squared error over the training set, without the adversarial perturbations applied), calculated at the end of every training epoch, with the perturbation applied over the inputs while bounded by $\bar{\delta}$. An inset highlights the initial training epochs, showing a slightly faster decrease in loss at small values of $\bar{\delta}$ ($\bar{\delta} = 0.01, 0.05$) compared to the $\bar{\delta} = 0$ (no perturbation) case.

Table 4: Final loss and accuracy on the training and test datasets for the trained model after 50 training epochs. The uncertainties are given by standard error (as compared to the standard deviation shown in Fig. (11).)

| Training | | Test | |
|---|---|---|---|
| Loss | Accuracy | Loss | Accuracy |
| 0.7322(73) | 0.9424(64) | 0.7245(78) | 0.9480(60) |

# 6 Classical dataset

The input data in the quantum dataset covered in the previous section are computationally-generated, exact quantum states. We also explore the effect of noise in cases where the input data may be "noisy"—where real-world errors and variation would be present between otherwise identical inputs, hence we test the QCNN on this classical MNIST dataset that uses real-world data.

In the noiseless case, a QCNN that is trained for 50 epochs decreases in loss as expected on the MNIST dataset, the mean and standard deviation of the loss at every epoch over 10 different sets of randomly generated initial parameters (using random seeds 0-9) are shown in Fig. (11) as follows.

There is greater variance in the final training results compared to the quantum dataset case, as this dataset is more representative of real-world noisy data. The parameter-loss landscape likely has more local minimas, and each set of initial parameters would define an initial position on this landscape. This leads to different sets of initial parameters descending into different local minima, from which we observe greater variance in final loss after training as compared to the quantum dataset.

For the trained models in Fig. (11), the accuracy (when using a cutoff of 0.5 to discriminate between positive and negative predictions) and loss will be:

We shall compare this case with the results from models with their training process subjected to the various sources and magnitudes of noise in the following subsections.

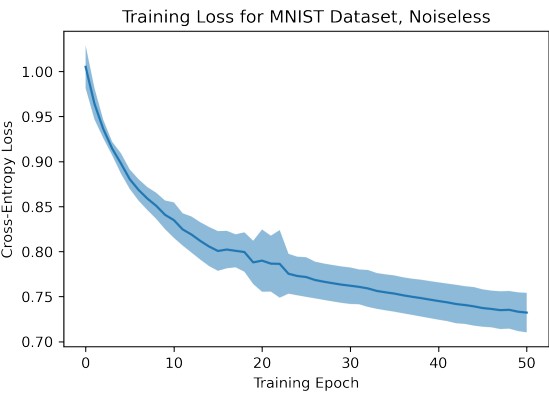

Figure 11: The training-loss plotted over 50 epochs of training. The Cross-Entropy Loss (as explained in section (11.2) over the training set is calculated at the end of every training epoch, after parameters are updated. The solid line indicates the mean loss over the 10 runs, and the shaded region indicates the standard deviation.

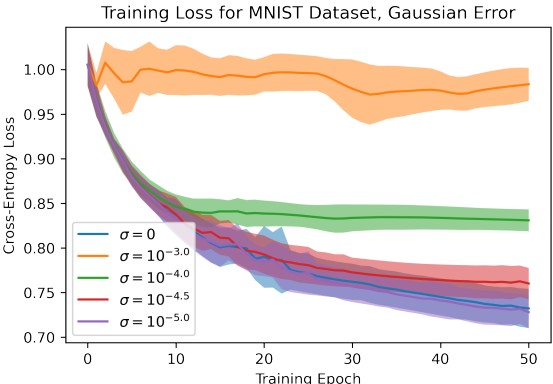

Figure 12: The loss (true cross-entropy loss over the training set, without any Gaussian noise applied), calculated at the end of every training epoch. Gaussian noise is applied with some constant standard deviation $\sigma$ throughout the gradient descent process for each case. The solid lines indicates the mean of the loss at each training epoch over the 10 repeated simulations, while the shaded regions indicate the standard deviation.

## 6.1 Gaussian noise

In a similar manner to the quantum dataset, we repeat the training process for 10 different sets of initial parameters generated from random seeds of $0-9$, and the training-loss curves with Gaussian noise of standard dev. $\sigma$ are shown in Fig. (12), and the loss on the training and test sets along with their standard error over the 10 runs are tabulated in table (5).

While the loss and accuracy for the $\sigma = 10^{-5}$ case are found to be lower than the noiseless case on average, they are well within standard errors of each other aand this indicates no statistically significant performance difference between the noiseless and Gaussian noise cases at this noise level. Though, the improvement in finalloss and accuracy at $\sigma = 10^{-5}$ may possibly be due to the Gaussian noise helping the gradient descent process to escape shallow local minima into deeper ones, as described in [20].

Similar to the quantum dataset case, we find that at $\sigma = 10^{-3}$, the Gaussian noise significantly affects the gradient descent process, preventing the loss from decreasing by any appreciable amount. The loss curves flatten out most likely due to the action of the bold-driver

Table 5: Final mean loss and mean accuracy on the training and test datasets for the trained model after 50 training epochs, with Gaussian noise at various values of $\sigma$ introduced during training. Uncertainties in brackets are given by the standard error.

| $\sigma$ | Training | | Test | |
|---|---|---|---|---|
| | Loss | Accuracy | Loss | Accuracy |
| 0 | 0.7322(73) | 0.9424(64) | 0.7245(78) | 0.9480(60) |
| 1e-3 | 0.9833(61) | 0.581(34) | 0.9825(64) | 0.582(37) |
| 1e-4 | 0.8309(41) | 0.9205(73) | 0.8271(37) | 0.9252(77) |
| 1e-4.5 | 0.7601(58) | 0.9402(68) | 0.7547(62) | 0.9454(73) |
| 1e-5 | 0.7281(61) | 0.9464(56) | 0.7210(64) | 0.9512(51) |

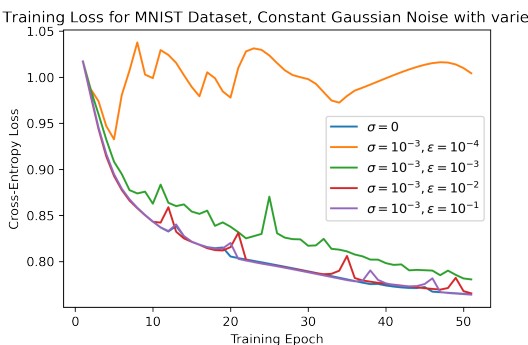

Figure 13: The true loss (without DPC applied) over the training set, calculated at the end of every training epoch, with DPC with various values of probability $p$ applied onto the output during the training process. All parameters are initialised to the same set of values using random seed $= 1$.

mechanism on the learning rate, similar to the quantum dataset as well.

In appendix (D), we also show that the gradients estimated at every epoch will also be normally distributed about the value which the parameter $c_i^{(l)}$ would be shifted by in the absence of noise, $\Delta c_i^{(l)} = \frac{\eta}{2\epsilon} \frac{1}{N} \sum_{k=1}^{N} \left\{ y^{(k)} \log_2 \left( \frac{h^{-(k)}}{h^{+(k)}} \right) + (1 - y^{(k)}) \log_2 \left( \frac{1 - h^{-(k)}}{1 - h^{+(k)}} \right) \right\}$, from eq. (29). We find that the variance has a lower bound proportional to:

$$\sigma^2_{\left( \Delta c_i^{(l)} \right)} \propto \left( \frac{\eta \sigma}{2(\ln 2)\epsilon} \right)^2 \frac{1}{N}. \tag{21}$$

Similar to the quantum dataset case, a large contributing factor to this variance is the small value of $\epsilon$—we can also reduce the effect of this random Gaussian error by increasing the value of the finite difference parameter $\epsilon$ here.

We can see that the effect of the Gaussian noise is also greatly mitigated, to the point that the $\epsilon = 10^{-2}$ curve is almost identical to the noiseless case. Once again, however, increasing $\epsilon$ will reduce the accuracy in estimating the gradient. The fact that this is especially evident when parameters are near minima is more visible here compared to the quantum dataset, where we see greater magnitudes in increases in loss close to the epochs where we see the noiseless case overshoot the minima.

## 6.2 Quantum depolarising channel

Fig. (14) shows how loss decreases throughout the training process for the classical dataset, with DPC applied with different magnitudes of $p$. Once again, the QCNN is initialised to the

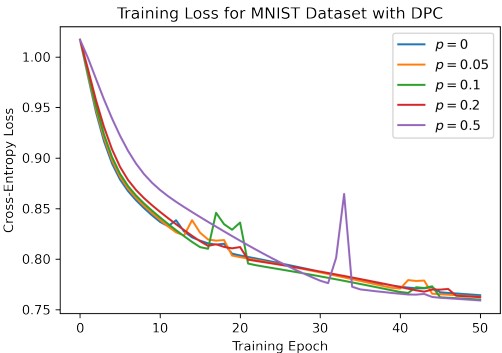

Figure 14: The true loss (without DPC applied) over the training set, calculated at the end of every training epoch, with DPC with various values of probability $p$ applied onto the output during the training process.

Table 6: The loss and accuracy (without DPC applied) on the training and test datasets for the trained model after 50 training epochs, with DPC at various values of $p$ introduced during training.

| $p$ | Training | | Test | |
|---|---|---|---|---|
| | Loss | Accuracy | Loss | Accuracy |
| 0 | 0.7643 | 0.9175 | 0.7615 | 0.9180 |
| 0.05 | 0.7627 | 0.9160 | 0.7598 | 0.9200 |
| 0.1 | 0.7600 | 0.9170 | 0.7570 | 0.9200 |
| 0.2 | 0.7624 | 0.9150 | 0.7595 | 0.9160 |
| 0.5 | 0.7591 | 0.9125 | 0.7561 | 0.9150 |

same set of parameters prior to the first training epoch using random seed $= 1$, and any evident differences are purely due to the effect of the DPC.

From the results, we notice two apparent effects of the DPC:

- Alike the quantum dataset case, the gradients decrease as $p$ is increased, although the relationship is not as clearly linear.

- Unlike the quantum dataset case, we observe a lower value of final loss on both the training and test sets in all the cases with DPC compared to the noiseless case. This reduction in loss does not correspond to a measurable change in accuracy.

To better characterize these observations, we quantify the effect of DPC on the training process. We derive how exactly the introduction of DPC will affect the gradient descent process in appendix (E).

To investigate the first point, we plot the ratio of the parameter update term under DPC compared to the noiseless equivalent case, for all gradients over the course of the 50 epochs for each case in Fig. (14). This corresponds to 351 parameters × 50 epochs in total, for each value of $p$.

Unlike the mean-squared error in the quantum dataset in Fig. (9), the mean of this ratio distribution in the logloss case does not correspond exactly to the value of $(1-p)$, this can also be seen from the derivation in appendix (E). Nonetheless, the difference from its equivalent noiseless value ($p = 0$) is greater as $p$ is increased, as predicted.

This is evident in the training-loss plots in Fig. (14)—the loss decreases at a slightly slower rate as $p$ is increased, close to the beginning.

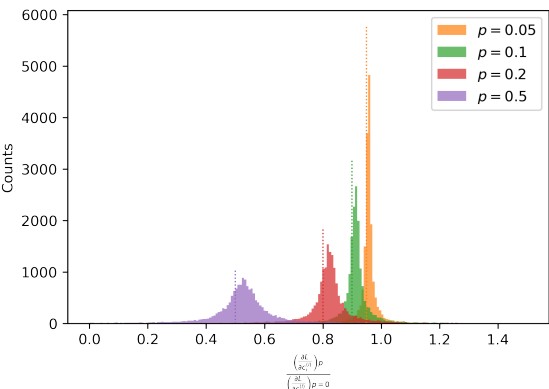

Figure 15: Histograms of the value of the parameter update term in eq. (E.3) divided by its equivalent noiseless value ($p = 0$), for every calculated gradient in the 50 training epochs for each value of $p$ shown in Fig. (14). We highlight the value (1 - $p$) along the $x$-axis for each histogram using a dotted vertical line of the same colour. The histogram is limited to values in the range [0, 1.5], which includes more than 90% of all 17550 values in all 4 cases.

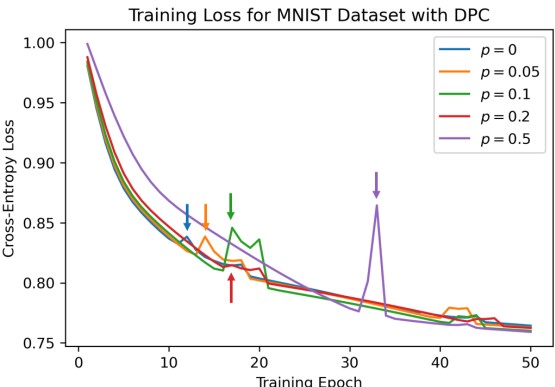

Figure 16: Identical Fig. to Fig.(14), with arrows added to highlight the first occurrences of parameters overshooting the minima and causing a significant increase in loss for each curve.

The second observed effect of the DPC is less straightforward—we expect that because of the lower values of estimated gradients, likely due to the fact that larger gradients are disproportionately affected by DPC in contrast to gradients with smaller values. To explain this counter-intuitive phenomenon, we begin with the observation that the first instances of overshooting the minima occur at a later training epoch as $p$ is increased, indicating that as p increases, the number of training epochs required for parameters to approach the minima increases, as shown in Fig. (16) below.

We conjecture that this delay in the overshooting of the minima for increasing $p$ is due to the fact that the specific parameters that are prone to this phenomenon scale more significantly with DPC than other parameters. We observe from Fig. (25) that at larger absolute values of $\log_2\left(\frac{a+x}{a-x}\right)$ on the vertical axis of the graph, a shift to $\log_2\left(\frac{a+x+\frac{p}{2(1-p)}}{a-x+\frac{p}{2(1-p)}}\right)$ leads to proportionally larger decreases, compared to the smaller absolute values of $\log_2\left(\frac{a+x}{a-x}\right)$.

Parameters that overshoot tend to have higher gradients, and consequently cause their parameter updates to be larger in value—the very reason why the parameter would be likely

to overshoot onto the other side of the minima. The higher gradients contain more terms at larger values of $\log_2\left(\frac{a+x}{a-x}\right)$, which in turn is affected by the DPC to a larger extent than terms with lower values.

This is supported by the fact that it takes three times the number of epochs for the $p = 0.5$ case to reach the minima, over the $p = 0$ case. Without even taking the increasing value of the learning rate into account, the parameters that overshoots would, on average, be shifted by 1/3 compared to the noiseless equivalent if it takes 3 times more iterative steps to reach the minima. If we take into account the increasing learning rate, this ratio will be even lower at 0.225, as estimated in appendix (F). In Fig. (15), we observe that the average ratio for all parameters tends to be around 0.53, a value higher than the ratio of 1/3 of the parameters that cause the overshooting.

The delay in the overshooting of the minima also delays the action of the bold-driver mechanism (which halves the learning rate when loss increases). This, in combination with the fact that most other parameters that do not cause overshooting are scaled less so by the DPC, allows most parameters to propagate a greater cumulative distance over many epochs of gradient descent.

As an example, the bold-driver mechanism halves the learning rate at epoch 12 and epoch 18 in the $p = 0$ case, which might lead to parameters in the $p = 0$ case covering a shorter cumulative distance in 31 epochs, corresponding to the observed lower loss in the $p = 0.5$ case. An estimated comparison between the cumulative distance in these 2 cases is shown in appendix (G).

Thus, DPC may alter training dynamics by delaying overshooting of the minima and the triggering of the bold-driver mechanism. This allows most parameters to descend a greater cumulative distance by the end of training, which corresponds to a lower final loss. However, this effect is conditional on the noiseless case overshooting the minima early in training. For other datasets or even other initial positions of the parameters (starting conditions) in this dataset, this might not occur at an early epoch. In such cases, the DPC cases show slower initial convergence compared to the noiseless case, as observed in the early ($< 10$) epochs of Fig. (14).

## 6.3 Adversarial training

We once again limit the perturbation by the adversary via bounding the 1-norm of the perturbative parameters by their absolute mean $\bar{\delta}$, $\frac{1}{3N}\left(\sum_{i=1}^{N}\sum_{j=1}^{3}|\delta_j^{(i)}|\right) < \bar{\delta}$, as described in section 3.3. For the QCNN in the classical dataset case, there are 8 qubits, and correspondingly a total of 24 of such parameters $\delta_j^{(i)}$, which are then trained by gradient ascent as described in section (12.3).

Fig. (8) shows how loss decreases throughout the training process when the perturbation is limited to various values of $\bar{\delta}$. We use the same set of initial parameters generated with a random seed of 1 throughout. The combination of gradient descent for both the perturbation parameters and the actual QCNN proves to be very computationally time-consuming. As such, we only test the low values of $\bar{\delta}$ that did not affect the training process too adversely in the quantum dataset case, and would be the most likely to produce interesting results.

We observe that increasing adversarial perturbation consistently leads to higher loss on both the training and test sets, and reduced accuracy, as expected.

While specific values of $\bar{\delta}$ may produce variations in training speed or accuracy, any such effects are outweighed by the increased computational time required due to the joint optimization process.

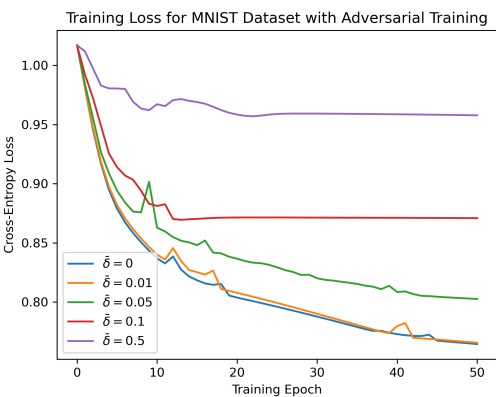

Figure 17: The loss (true mean-squared error over the training set, without the adversarial perturbations applied), calculated at the end of every training epoch, with the perturbation applied over the inputs while bounded by $\bar{\delta}$.

Table 7: Final loss and accuracy on the training and test datasets by the trained model after 50 training epochs, with the perturbation applied over the inputs bounded by $\bar{\delta}$.

| $\bar{\delta}$ | Training | | Test | |
|---|---|---|---|---|
| | Loss | Accuracy | Loss | Accuracy |
| 0 | 0.7643 | 0.9175 | 0.7615 | 0.9180 |
| 0.01 | 0.7654 | 0.9160 | 0.7625 | 0.9150 |
| 0.05 | 0.8024 | 0.8895 | 0.7993 | 0.8990 |
| 0.1 | 0.8708 | 0.8760 | 0.8656 | 0.8830 |
| 0.5 | 0.9578 | 0.7545 | 0.9553 | 0.7740 |

## 6.4 Comparison with other machine learning models

Classical machine learning algorithms can achieve accuracy of more than 99% on the MNIST dataset. A list of top-performing models with code and articles linked are available at [28]. However, note that our QCNN is trained with only a small subset of the original MNIST dataset—only 2000 of the original 60000 training examples were used. In addition, many of the classical models have more trainable parameters than the QCNN we used. For reference, one example [29] contains 1,514,187 parameters, more than 3 magnitudes more than our model, which has only 351 parameters.

A more comparable example that uses an alternative architecture of QCNN [30] is able to achieve an average accuracy of 0.948 in distinguishing between the digits "1" and "8", when trained on 5000 examples while using 46 trainable parameters. This accuracy is similar in value to the accuracy of the QCNN in our work in most cases.

# 7 General discussions over both datasets

## 7.1 Gaussian noise

We find that the effect of Gaussian noise on the gradient descent process in this work is consistent with some other numerical results on other quantum machine learning models, particularly Fig. 4a from [31].

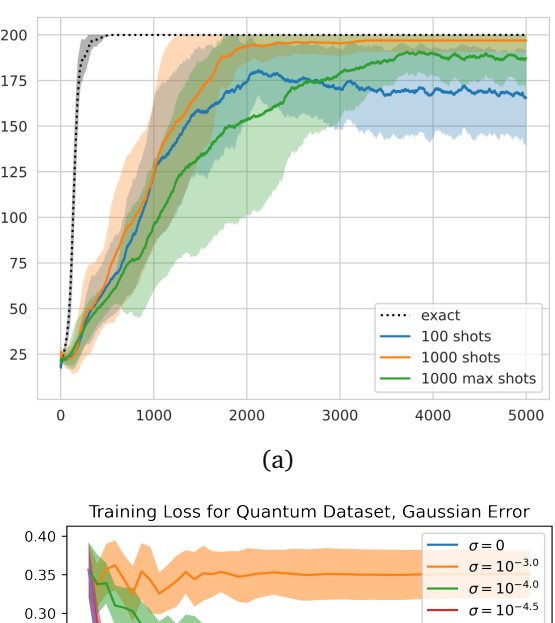

(a)

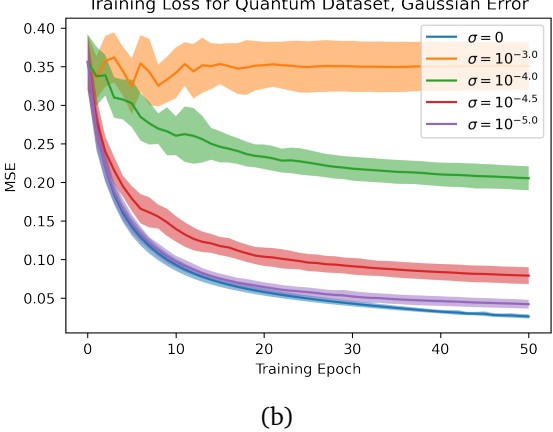

(b)

Figure 18: (a) Fig. 4a from [31]. The score (*y*-axis) at every training epoch (*x*-axis) for a quantum reinforcement learning model from [31]—higher score represents better performance. The number of shots represent the number of measurements made on the output—a higher number of shots would indicate a lower value of standard error, and is likely to be closer to the actual output. (b) Fig. (6) showing the effect of Gaussian noise on our quantum dataset, for comparison.

In Fig. (18), we see that taking more shots (measurements) of the output would improve both the average performance at the end, as well as a faster increase in score each epoch. As a higher number of shots would indicate a lower value of standard error, this is similar to applying a lower value of Gaussian noise onto the output in our case. This is consistent with our results, in that applying a lower Gaussian noise would generally increase the rate of loss decreasing per epoch, and also leading to a lower final loss.

In both our datasets, we were able that show that a Gaussian Noise applied onto the output of the QCNN would cause the parameter update term in the gradient descent process to be normally distributed about its noiseless value. If we take the expectation value of the parameter update term under Gaussian noise, instead of randomly generating a random normal variable, one can recover its equivalent noiseless training-loss curve, as the training process will be identical to the noiseless case.

While we have proven this to be true for both mean-squared error (in the quantum dataset case) and binary cross-entropy loss (in the MNIST dataset case), it may not necessarily hold true for all forms of loss functions. It may be possible that there may exist some loss functions where if used in an experimental implementation of a quantum machine learning cir-

cuit, would cause the gradient descent process to be biased from experimental errors—the expectation value of the parameter update term may not necessarily be the same as the value in an ideal noiseless case. As such, when using loss functions other than mean-squared error or cross-entropy loss, it may be advisable to investigate whether the parameter update term is truly distributed about the noiseless scenario, whether through analytic derivation or numerical simulation as was done in this work.

### 7.2 Quantum depolarising channel

The effect of DPC we have simulated in the QCNN appears to be consistent with the effect of similar simulated noise in other quantum machine learning models. We cite one result [32] of applying the amplitude-damping channel on a single-qubit classifier, applied to the MNIST task as well. We are able to see parallels between the results in this work, and our DPC results on both our datasets. Specifically, we observe that, as in [32], the loss decreases at a lower rate as the noise parameter increases—where $\lambda$ in [32] plays an equivalent role to $p$ in our work, as shown in Figures 9 and 11.

To summarise our results over both datasets, a DPC of probability $p$ applied onto the outputs would have the following effects on the gradient descent process:

- Most parameter updates are scaled by some value proportional to the magnitude of $p$, on average. Therefore, the larger the value of $p$, the slower the rate of decrease in loss.

- When mean-squared error is used, at low values of loss, the parameter updates tend to zero in the noiseless case due to $h^{(k)} \approx y^{(k)}$. However, the factor of $(1-p)$ applied onto predicted values causes parameters to continue updating in the DPC cases even at low values of loss, when $p$ is significant. This increases incidences of overshooting the minima.

- In cases where the noiseless case (without DPC) will overshoot the minima early and lead to an increase in loss, the lower rate of parameter updates in the cases with DPC might delay this. This might help the loss to eventually be lower than the noiseless case, when DPC is applied. This was only shown for the MNIST dataset when cross-entropy loss is used.

Though the noise does not affect the training process much, should one want to mitigate this source of error, one approach can be adjusting the predicted value to account for the change caused by the DPC. If an experimental setup is found to have decoherence at some specific value of $p$, the true predicted value $h$ be found from the measured predicted value $h'$ by $h = \frac{1}{1-p}h'$ when using expectation value (as in the case of the quantum dataset). In the case of probability of measuring $|+\rangle$, the same can be achieved by $h = \frac{1}{1-p}h' - \frac{p}{2(1-p)}$.

## 8  Conclusions

This work aimed to quantify the effects of three types of noise—Gaussian noise, depolarizing channel noise (DPC), and systematic errors during initial state preparation, on the operation of a quantum convolutional neural network (QCNN). We analyzed how each noise type influences training dynamics, convergence behavior, and final model performance, with the goal of identifying strategies to mitigate detrimental effects in experimental implementations.

One specific observation may warrant further investigation—DPC noise modifies training time characteristics by slowing convergence and delaying learning rate adjustments triggered by the bold-driver mechanism. In some cases, this results in lower final loss values. Further

study may clarify whether this effect can be systematically utilized to optimize training times under certain conditions.

For the quantum dataset, where inputs consist of exact wavefunctions, noise consistently reduced performance. This aligns with the expectation that noise is generally detrimental when applied to ideal quantum data, contrasting with its role in classical machine learning where it is often introduced to improve model generalizability.

Applying quantum machine learning to quantum datasets that contain real-world experimental noise, or alternative quantum machine learning models, may be a productive direction for further quantification. Examples include many-body quantum state tomography using experimental data, as previously studied with classical neural networks on simulated data [33], or using QCNNs as error-correction decoders that learn characteristic noise profiles of specific quantum computers, as suggested in [34].

In sections (5.1) and (6.1), we quantified how Gaussian noise of varying magnitudes affects parameter updates during gradient descent. We also observed how adjusting the finite-difference parameter $\epsilon$ used in gradient estimation changes the sensitivity of gradient descent to Gaussian noise. For experimental implementations, tuning $\epsilon$ may help balance gradient estimation accuracy with noise robustness.

Lastly, our findings are consistent with results from other quantum machine learning models. For example, in [32], the amplitude-damping channel applied to a single-qubit classifier on MNIST showed similar behavior, with loss decreasing at a slower rate as the noise parameter $\lambda$ increased. Extending this quantification approach to other forms of quantum error, such as amplitude-damping channels relevant to atomic systems, would be a natural continuation of this work.

## Methods & implementation

To ensure the reproducibility of our data, we detail the steps and implementation for the simulation of the QCNN, the creation and usage of the datasets the QCNN will be applied on, as well as how the various sources of noise are simulated.

All quantum circuits and simulated noise in this work can be implemented in Python on the Qiskit [22] or QIBO [35] quantum circuit simulation platforms.

## 9 QCNN

Notable implementation decisions and changes to the original QCNN architecture will be described in the following subsections. For ease of reference, we once again show a copy of the QCNN architecture from Figure (1) here.

### 9.1 Initialisation of parameters

Different sets of initial parameters may possibly lead to different local minima during the gradient descent process, similar to how the dynamics of a physical system depend on the initial conditions. To compare the effect of only varying the magnitude of the applied noise, we need to initialize the parameters to the same set of values.

Considering this, and to ensure the reproducibility of our results, the parameters for the QCNN (**c**) are initialized to random values as generated through the random.RandomState random number generator from NumPy [36] with specified random seeds in each respective section in chapter 3. Using the same random seed would generate the same set of random

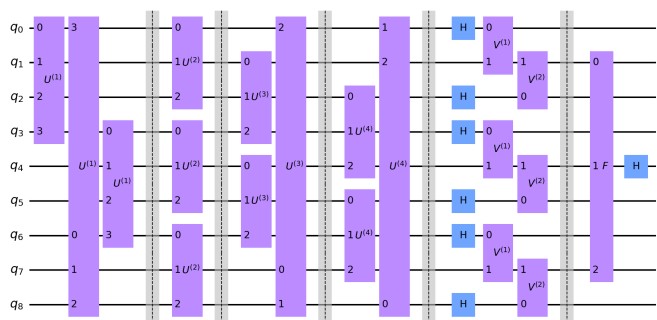

Figure 19: Example graphic representation of the QCNN architecture, with $N_p = 3$ and 9 total qubits. Qubits are evolved through the circuit from left to right, though unitary operators represented by purple blocks. The numbers aligned along the left of each block represent the qubits on which they act. The blue (H) blocks are Hadamard gates, which transform the qubits into the $x$-basis. The expectation value of the qubit $q_4$ at the end of the circuit is taken as the output of the QCNN in this case.

values, which leads to the same set of initial parameters for reproducibility.

## 9.2 Retrieval of output

The output of the QCNN is given by the expectation value of a single qubit as described in eq. (4). Instead of simulating the repeated measurements of this qubit as is usually done for Qiskit circuits, we take the statevector at the end of the QCNN and obtain the probabilities of each possible basis state. The probability of measuring the measured qubit in the $|+\rangle$ state is taken as the sum of the probabilities of all basis states where the qubit is in the $|+\rangle$ state.

This proves to be more efficient compared to simulating repeated runs of the circuit to obtain the fraction of qubits measured as $|+\rangle$, and is also able to provide a precise value of the expectation value of measuring the $|+\rangle$ state for simulating gaussian and DPC noise upon.

## 9.3 Changes to pooling layer

In the interest of computational efficiency, we similarly do not simulate the measurement of the pooled qubits in the pooling layer. Instead, we utilize controlled unitary operators, which are shown as $V^{(1)}$ and $V^{(2)}$ in Figure (19). These can be represented as the following matrix acting upon a 2-qubit statevector $|\psi\rangle = a_1 |00\rangle + a_2 |01\rangle + a_3 |10\rangle + a_4 |11\rangle$:

$$V^{(i)} |\psi\rangle = \begin{pmatrix} I_2 & 0 \\ 0 & U'_c \end{pmatrix} \begin{pmatrix} a_1 \\ a_2 \\ a_3 \\ a_4 \end{pmatrix}. \tag{22}$$

With $U'_c$ being a $2 \times 2$ unitary matrix, which acts on the second qubit (labelled as 1 in Figure (19) for each $V^{(i)}$ operator) only if the first qubit (labelled as 0 in Figure (19)) is in a $|1\rangle$ state, otherwise the second qubit is left unchanged. The unitary matrix will be generated on the trainable parameters of the QCNN. For each of the $N_p - 1$ pooled qubits, there will be a corresponding $U_c$ that acts on the one remaining qubit depending on the value of the pooled qubit.

Understandably, this is a unitary process compared to the original non-unitary process which involves measurement of the pooled qubits. As mentioned at the end of section (1.1), these non-unitary measurement processes are what introduce non-linearities to the QCNN,

so this form of implementation of the pooling layer may reduce the QCNN's ability to model non-linear relationships - the only non-linear process remaining would be the measurement of the single measured qubit at the end.

### 9.4 Training process

The trainable parameters of the QCNN are the coefficients $c_i^{(l)}$ of the generalised Gell-Mann matrices, used to generate the unitary operators. In the gradient process, the partial derivative of the loss function with respect to each parameter $c_i^{(l)}$ is required. This can be approximated by the finite difference method:

$$\frac{\partial L(h[|\psi\rangle_{\text{in}}^{(1,2,...N)}; \mathbf{c}], y^{(1,2,...,N)})}{\partial c_i^{(l)}} \approx \frac{1}{2\epsilon} \left( L(h^+, y^{(1,2,...,N)}) - L(h^-, y^{(1,2,...,N)}) \right). \quad (23)$$

$$\text{With } h^+ = h[|\psi\rangle_{\text{in}}^{(1,2,...N)}|(c_1^{(1)}, c_2^{(1)}, ..., c_i^{(l)} + \epsilon, ...)],$$

$$\text{and } h^- = h[|\psi\rangle_{\text{in}}^{(1,2,...N)}|(c_1^{(1)}, c_2^{(1)}, ..., c_i^{(l)} - \epsilon, ...)].$$

The variable $h^+$ would be the output of the QCNN when the parameter $c_i^{(l)}$ is increased by $\epsilon$, and $h^-$ would be the output when the parameter $c_i^{(l)}$ is decreased by $\epsilon$.

Again, it is understood that the loss function is to be summed over all pairs of input wavefunctions and their corresponding target labels $(|\psi\rangle_{\text{in}}^{(1)}, y^{(1)})$, $(|\psi\rangle_{\text{in}}^{(2)}, y^{(2)})$,... ,$(|\psi\rangle_{\text{in}}^{(N)}, y^{(N)})$, for the $N$ training examples in the training dataset. Shifting some parameter $c_i^{(l)}$ by a small finite value $\epsilon$ would change its corresponding unitary operator accordingly, which changes the output of the QCNN and thus the loss function. The finite difference method compares the value of the loss function at $c_i^{(l)} + \epsilon$ and $c_i^{(l)} - \epsilon$, while holding all other parameters constant, to estimate the partial derivative of the loss function with respect to $c_i^{(l)}$.

As prescribed by the gradient descent process, each parameter is updated according to its corresponding partial derivative of the loss function at every training epoch by:

$$c_i^{(l)} \leftarrow c_i^{(l)} - \eta \frac{\partial L(h[|\psi\rangle_{\text{in}}^{(1,2,...N)}; \mathbf{c}], y^{(1,2,...,N)})}{\partial c_i^{(l)}}. \quad (24)$$

The coefficient $\eta$ is the learning rate, which is controlled by the bold-driver mechanism [37]—$\eta$ is increased by 5% if the loss in a training epoch decreases compared to the previous epoch, and decreased by 50% if it does not.

We use a value of $\epsilon = 10^{-4}$ and an initial value of $\eta = 1$ in all of our tests unless specified otherwise. For all tests, we will run 50 training epochs—iteratively repeating the above gradient process 50 times.

## 10 Quantum dataset

The code used to generate the quantum dataset is implemented in libraries QuTip [38] and TenPy [39].

The inputs $|\psi\rangle_{\text{in}}$ to the QCNN for the QPR task would be the ground state wavefunctions to the Hamiltonian eq. (6) at various values of $h_1/J$ and $h_2/J$. We generate these by diagonalizing the Hamiltonian for a system of $N = 9$ spin sites, at 51 evenly spaced points in $h_1 \in [0, 1.6]$ and 51 evenly spaced points of $h_2 \in [-1.6, 1.6]$ at $J = 1$. Note that there are two degenerate ground states when $h_1$ is exactly 0, for all values of $h_2$. To lift this degeneracy, we use $h_1 = 10^{-6}$ when deriving any values associated with $h_1 = 0$. There are 4 degenerate

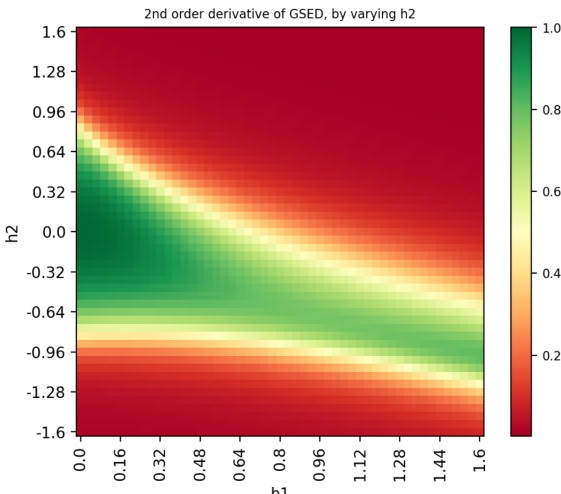

Figure 20: The value of the estimated second order derivative of the ground state energy density at various discrete points $(h_1, h_2)$.

$h_1 = 0, h_2 = 0$ that cannot be avoided with this approach, but we shall keep it in the dataset as a "noisy" point that might help against overfitting.

As boundary effects would be present when simulating a finite chain length, the behaviour (such as phase transitions) of a small number of spin sites may differ from that of a system with a larger number of spin sites. However, using larger number of spin sites would exponentially increase the computation time of the QCNN (in which hilbert space grows by a factor of two for every additional qubit.) We find that the generated wavefunctions for 9 spin sites to be sufficient in allowing the QCNN to predict the SPT phase reliably, within reasonable computation time.

As mentioned in section (2.1), we use the second order derivative of the ground state energy density to estimate the SPT phase, which produces the dataset. We estimate this numerically using the finite difference method along $h_2$ using a small finite value $\epsilon$. At a particular point $(h_1, h_2)$, this will be:

$$y^{(h_1, h_2)} = \frac{1}{\epsilon^2} \left( E_{|\psi(h_1, h_2 + \epsilon)\rangle} - 2E_{|\psi(h_1, h_2)\rangle} + E_{|\psi(h_1, h_2 - \epsilon)\rangle} \right). \tag{25}$$

Here, $E_{|\psi(h_1, h_2)\rangle}$ is the energy of the ground state wavefunction $|\psi(h_1, h_2)\rangle$ at $(h_1, h_2)$. Using a value of $\epsilon = 10^{-3}$, we are able to obtain a good estimate of the SPT phase, shown in Figure (20) below.

For the quantum dataset, we opt to follow the original structure in [21] as well, having a using $L = 9$ qubits and pooling size of $N_p = 3$ qubits. This pooling size is used as the original Hamiltonian of the spin chain consists of a term across 3 sites. The structure of the QCNN is depicted in Figure (19), and there is 513 trainable parameters $c_i^{(l)}$ in total throughout the circuit.

## 10.1 Training and test datasets

We use the 51 data points along the $h_2 = 0$ row as the training set, as it contains a well-varied range of target values from 0 to 1, representative of the rest of the dataset. The entire dataset—all $51 \times 51$ discrete points for $h_1 \in [0, 1.6]$ and $h_2 \in [-1.6, 1.6]$ in the parameter space described prior is then used as the test set.

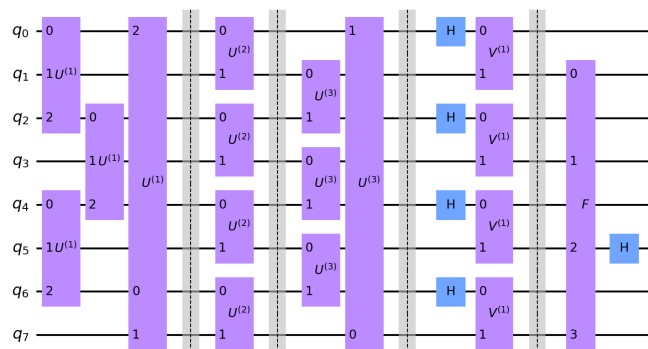

Figure 21: Graphic representation of the QCNN structure we use for the MNIST dataset, with description similar to Figure (19). The probabiltiy of the qubit $q_5$ measured as the $|+\rangle$ state is taken as the output of the QCNN in this case.

## 10.2 Loss function

For the quantum dataset, we opt to use the mean squared loss (MSE) as the loss function for this task, as we want our QCNN to predict the second order derivative of the ground state energy density, a continuous value.

$$L(h[|\psi\rangle_{\text{in}}^{(1,2,...N)}; \mathbf{c}], y^{(1,2,...,N)}) = \frac{1}{N}\sum_{i=1}^{N}\left\{h[|\psi\rangle_{\text{in}}^{(i)}; \mathbf{c}] - y^{(i)}\right\}^2 . \tag{26}$$

This will be the loss function that will be used in the place of $L$ in equations under section (9.4), for any tests involving the quantum dataset. The gradient descent process will therefore update each parameter by:

$$c_i^{(l)} \leftarrow c_i^{(l)} - \frac{\eta}{2\epsilon}\left[\frac{1}{N}\sum_{k=1}^{N}\left\{h^{+(k)} - y^{(k)}\right\}^2 - \frac{1}{N}\sum_{k=1}^{N}\left\{h^{-(k)} - y^{(k)}\right\}^2\right]. \tag{27}$$

With $h^{+(k)}$ once again denoting the output value for the $k^{th}$ input when the parameter $c_i^{(l)}$ is increased by a small finite value $\epsilon$, and $h^{-(k)}$ the output value when the parameter $c_i^{(l)}$ is decreased by $\epsilon$, as per the finite difference method as described in eq. (23). The second term is the value by which $c_i^{(l)}$ will be shifted/updated by, and this is termed the "parameter update term".

# 11  Classical dataset

From the original 28 × 28 MNIST images, we resize them to 16 × 16 using the tf.image.resize function from Tensorflow [40]. The 256 values in such a 16 × 16 image can be encoded in the amplitudes of the $2^8$ possible computational basis states of exactly 8 qubits. We therefore use a QCNN of $L = 8$ qubits with $N_p = 2$ pooling size, which is shown in Figure (21) below; there is 351 trainable parameters $c_i^{(l)}$ in total throughout the circuit.

## 11.1  Training and test datasets

The original MNIST dataset contains approximately 10000 images for each digit labelled from "0" to "9" in the training set, and approximately 1000 images for each digit in the test set.

We downsample this original dataset to create balanced training and test sets (equal number of positively-labelled and negatively-labelled data), and to reduce computation time. We randomly sample 1000 images labelled as the digit "1" for our positive data ($y^{(i)} = 1$) and 1000 images labelled as digits other than "1" as the negative data ($y^{(i)} = 0$) to comprise our training dataset. The test set similarly uses 500 images labelled as "1" and 500 images labelled as any digit other than "1" from the original test set. The random.RandomState random number generator from NumPy with a specified random seed of 1 is used to generate this random sample, and one should be able to obtain the exact same set of data we used by doing so.

## 11.2 Loss function

For this dataset, we opt to use the binary cross-entropy loss [24], or logloss, which is commonly used in classification tasks where the target values take on values only two possible values, such as zeroes or ones, $y^{(i)} = \{0, 1\}$.

$$
\begin{aligned}
L(h[&|\psi\rangle_{\text{in}}^{(1,2,...N)}; \mathbf{c}], y^{(1,2,...,N)}) \\
&= -\frac{1}{N} \sum_{k=1}^{N} \left\{ y^{(k)} \log_2 \left( h[|\psi\rangle_{\text{in}}^{(k)}; \mathbf{c}] \right) + (1 - y^{(k)}) \log_2 \left( 1 - h[|\psi\rangle_{\text{in}}^{(k)}; \mathbf{c}] \right) \right\}.
\end{aligned} \tag{28}
$$

This loss function is derived from the concept of entropy in information theory, and it measures the amount of informational value relevant to the classification task, that the model is able to produce from the inputs. It is a continuous function differentiable with respect to $h[|\psi\rangle_{\text{in}}^{(k)}; \mathbf{c}]$ to at least first order even when the target values $y^{(k)}$ are discrete, which is important as the partial derivative of the loss function need to be computed for each trainable parameter. In this context, the output of the NN $h[|\psi\rangle_{\text{in}}^{(k)}; \mathbf{c}]$ from an input $|\psi\rangle_{\text{in}}^{(k)}$ can be interpreted to be the probability of this input being classified as the object whose target values are labelled as $y^{(k)} = 1$.

As can be seen, the $L$ above has a minimum value of 0, which is achieved when the predicted values all match the target values $h[|\psi\rangle_{\text{in}}^{(i)}; \mathbf{c}] = y^{(i)}$ for all $i$, and a maximum value approaching $-\infty$ when any predicted value are opposite the target value, which heavily penalises incorrect predictions. Using this loss function presents two issues with the original implementation of the QCNN.

- The predicted value $h[|\psi\rangle_{\text{in}}^{(i)}; \mathbf{c}]$ is taken as the expectation value of one qubit, which has range [-1,1]. Using this predicted value in the logarithm $\log_2$ of the loss function above would cause it to be undefined whenever $h[|\psi\rangle_{\text{in}}^{(i)}; \mathbf{c}] < 0$.

- The loss function is also undefined whenever $h[|\psi\rangle_{\text{in}}^{(i)}; \mathbf{c}] = 0$ or 1.

To remedy this issue, we make the following two changes to the original QCNN.

- We instead use the probability of measuring the state $|+\rangle$ rather than the expectation value as the output value of the QCNN, i.e. $h[|\psi\rangle_{\text{in}}^{(i)}; \mathbf{c}] = |\alpha|^2$ instead of $2|\alpha|^2 - 1$ from eq. (5).

- We also offset the value by a small value $10^{-6}$ whenever $h[|\psi\rangle_{\text{in}}^{(i)}; \mathbf{c}] = 0$ or 1 to avoid the singularity of the loss function at these values.

Finally, with this loss function, the gradient descent process will update each parameter every epoch by:

$$
c_i^{(l)} \leftarrow c_i^{(l)} - \frac{\eta}{2\epsilon} \left[ \frac{1}{N} \sum_{k=1}^{N} \left\{ y^{(k)} \log_2 \left( \frac{h^{-(k)}}{h^{+(k)}} \right) + (1 - y^{(k)}) \log_2 \left( \frac{1 - h^{-(k)}}{1 - h^{+(k)}} \right) \right\} \right]. \tag{29}
$$

The notation as described in eq. (23) is used, once again. The second term is the value by which $c_i^{(l)}$ will be shifted/updated by, and this is termed the "parameter update term".

# 12 Implementations of noise

## 12.1 Gaussian noise

As the probability $|\alpha|^2$ when measuring a single-qubit state $|\psi\rangle = \alpha\,|+\rangle + \beta\,|-\rangle$ is limited to $[0, 1]$, we apply a truncated Gaussian distribution with variance $\sigma^2$ unto $|\alpha|^2$. This distribution has the probability density function:

$$f(x; \mu, \sigma, a, b) = \frac{1}{\sigma} \frac{\phi\left(\frac{x-\mu}{\sigma}\right)}{\Phi\left(\frac{b-\mu}{\sigma}\right) - \Phi\left(\frac{a-\mu}{\sigma}\right)}, \tag{30}$$

$$\text{with } \phi(x) = \frac{1}{\sqrt{2\pi}} \exp(-x^2/2), \Phi(x) = \frac{1}{2}\left(1 + \text{erf}(x/\sqrt{2})\right).$$

Here, $\phi$ is the probability density function of the standard normal distribution, $\Phi$ its cumulative density function, and erf is the error function. In our case, the limiting values $a, b$ are specified as $0, 1$ respectively, and the mean $\mu$ is taken to be the original probability $|\alpha|^2$. At small values of $\sigma$ and if the mean $\mu$ is sufficiently far away from the limits $a, b$, this reduces to a normal distribution. This is indeed the case for most of our usage in this work.

The value of $|\alpha|^2$ used in the calculation of the expectation value of the measured qubit, and consequently the output of the QCNN, will be a randomly-generated value that follows the distribution above. As we will be investigating the effect of various magnitudes of noise, we shall test the QCNN at various values of $\sigma$ in chapter 3.

This randomness in the predicted values will cause randomness in the gradient process as well, since the loss functions and consequently the gradient is dependent on the predicted value. This may potentially cause parameters to be updated in the opposite direction of the gradient. When this occurs, the loss may increase and trigger the bold-driver mechanism to halve the learning rate $\eta$.

Halving the learning rate in the bold-driver method seek to remedy the "overshooting the minima" phenomenon indicative of an overly high learning rate. As this would not be helpful when it is triggered by the aforementioned random fluctuations in loss due to Gaussian noise, we relax the threshold to halve the learning rate only when the increase in loss is more than $4\sigma$ over the previous epoch. This specific value of $4\sigma$ is chosen, since the value by which parameters will be updated by the gradient descent process will be a random normal variable with standard deviation proportional to $4\sigma$ for the quantum dataset case, when Gaussian noise of with variance $\sigma^2$ is introduced. This is derived in section (5.1) and Appendix (A).

Due to the stochastic nature of this Gaussian noise, we also repeat the training process for 10 different sets of initial parameters generated from random seeds of $0-9$. This is done as we would like to confirm that there exists some benefit on average (if any) from applying Gaussian noise, since for single cases, any benefit may be merely due to random chance.

## 12.2 Quantum depolarising channel

We apply the transformation by the DPC as specified by eq. (12) on the reduced density matrix of the measured qubit at the end of the QCNN.

In a real experimental implementation, it is possible for the state to be affected by decoherence throughout the circuit, prior to passing through each of the unitary operators. Given that

a unitary operator $U_1$ would act on a density matrix $\rho$ by $U_1 \rho U_1^\dagger$, the operation of a unitary operator on a density matrix after a DPC of probability $p_1$ can be expressed as:

$$U_1 \left( p_1 \frac{I}{2} + (1-p_1)\rho \right) U_1^\dagger = \left( p_1 \frac{I}{2} + (1-p_1) U_1 \rho U_1^\dagger \right). \tag{31}$$

If it undergoes a DPC with probability $p_2$ afterwards,

$$p_1 \frac{I}{2} + (1-p_1)U_1 \rho U_1^\dagger \rightarrow p_2 \frac{I}{2} + (1-p_2)\left( p_1 \frac{I}{2} + (1-p_1)U_1 \rho U_1^\dagger \right)$$
$$= (p_2 - p_2 p_1 + p_1)\frac{I}{2} + (1 - p_2 + p_2 p_1 - p_1)U_1 \rho U_1^\dagger. \tag{32}$$

We can express $(p_2 - p_2 p_1 + p_1)$ as a new probability $p'$, and the density matrix above would be equivalent to the matrix $U\rho U^\dagger$ under the effect of DPC with probability $p'$. Applying another unitary operator $U_2$ as per eq. (31) would then result in $\left( p' \frac{I}{2} + (1-p')U_2 U_1 \rho U_1^\dagger U_2^\dagger \right)$. Repeatedly applying this process would result in some state:

$$\left( p_{\text{final}} \frac{I}{2} + (1-p_{\text{final}})U_n...U_2 U_1 \rho U_1^\dagger U_2^\dagger..U_n^\dagger \right). \tag{33}$$

Therefore, the process of successively applying unitary operators on a state which undergoes decoherence represented by a DPC with some probability $p_i$ in between each unitary, can be shown to be equivalent to applying a DPC with some probability $p_{\text{final}}$ on the density matrix of the final state. As our QCNN consists of purely unitary operators, we can model such decoherence over the circuit by just applying the DPC on the final state, prior to measurement of the one measured qubit. If the decoherence by each layer is at a constant $\bar{p}$, the $p_{\text{final}}$ after $N$ layers is:

$$p_{\text{final}} = 1 - (1 - \bar{p})^N. \tag{34}$$

However, this method of applying the DPC onto the final state may not be able to accurately model the effect of decoherence if the circuit involves non-unitary processes in the middle, such as the original implementation of the pooling layer, which involves measurements of the pooled qubits. Application of any non-unitary operators would not be canceled out when multiplied around the $I/2$ term.

The training process would be deterministic under this implementation of the DPC noise—with the same set of initial parameters at some constant specified $p$, the training process would be identical over repeated runs, and lead to the same final result.

## 12.3 Adversarial training

As described in section (3.3), we first perturb the input state by individual unitary operators on each qubit.

Each perturbation unitary $U_\delta^{(i)}$ is generated from some linear combination of its generators, which in this case will be the Pauli matrices. The tensor product of these single-qubit operators for all $L$ qubits determines an overall perturbation operator $U_\delta$ on input states $|\psi\rangle_{\text{in}}$ before the QCNN.

$$U_\delta^{(i)} = \exp\left( -i(\delta_1^{(i)} X + \delta_2^{(i)} Y + \delta_3^{(i)} Z) \right),$$
$$U_\delta = U_\delta^{(1)} \otimes U_\delta^{(2)} ... \otimes U_\delta^{(L)},$$
$$|\psi\rangle_{\text{in}} \mapsto U_\delta |\psi\rangle_{\text{in}}.$$

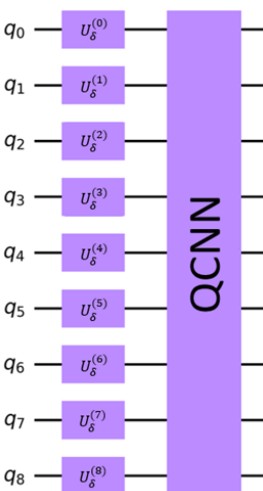

Figure 22: Circuit diagram showing the placements of the perturbation operators prior to the QCNN.

To find the set of parameters $\delta_j^{(i)}$ that maximizes the loss function, as per eq. (15), 10 training epochs of gradient "ascent" is performed prior to every training epoch of the main QCNN. Instead of updating these $\delta_j^{(i)}$ in the direction of a local minima, $\delta_j^{(i)}$ are updated in the direction of some local maxima:

$$\delta_j^{(i)} \leftarrow \delta_j^{(i)} + \eta_\delta \frac{\partial L(h[U_\delta |\psi\rangle_{\text{in}}^{(1,2,\dots N)}; \mathbf{c}], y^{(1,2,\dots,N)})}{\partial \delta_j^{(i)}} \, . \tag{35}$$

We use a constant learning rate $\eta_\delta = 1$. To limit this perturbation to some value $\bar{\delta}$ as per eq. (17), we "normalise" the vector of parameters after every such training epoch, by its 1-norm, if the 1-norm exceeds the value of $\bar{\delta}$:

$$\left(\delta_1^{(1)}, \delta_2^{(1)}, \dots \delta_3^{(L)}\right) \leftarrow \bar{\delta} \frac{\left(\delta_1^{(1)}, \delta_2^{(1)}, \dots \delta_3^{(L)}\right)}{\left(\sum_{i=1}^N \sum_{j=1}^3 |\delta_j^{(i)}|\right)} \, . \tag{36}$$

In this manner, the parameters that have higher gradients (higher value of $\frac{\partial L(h[U_\delta |\psi\rangle_{\text{in}}^{(1,2,\dots N)}; \mathbf{c}], y^{(1,2,\dots,N)})}{\partial \delta_j^{(i)}}$) can still be increased at the cost of decreasing other parameters with lower gradients, which increases the loss even while limiting the 1-norm of parameters to $\bar{\delta}$.

Though adding such perturbations may increase the loss over subsequent training epochs and trigger the bold-driver method alike the Gaussian noise case, we expect that any increase may be overshadowed by the decrease in loss from the gradient descent of the main parameters in the QCNN, especially when the perturbative parameters are limited by small values of $\bar{\delta}$.

Alike the main parameters in the QCNN, the final local maxima that these parameters ascend to would also depend on their initial positions in parameter space, so we also initialise these parameters to consistent, repeatable sets of values using specified random seeds.

# Acknowledgments

**Funding information** This work is supported by the Singapore Ministry of Education (MOE) and the Singapore National Research Foundation (NRF).

# A Effect of gaussian noise on parameter update with MSE loss

We begin from eq. (27), replacing the hypotheses from the QCNN $h^{+(k)}, h^{-(k)}$ by their corresponding normal distributions due to the introduction of Gaussian noise (eq. 10).

$$c_i^{(l)} \leftarrow c_i^{(l)} - \frac{\eta}{2\epsilon} \left[ \frac{1}{N} \sum_{k=1}^{N} \left\{ \mathcal{N}\left(h^{+(k)}, 4\sigma^2\right) - y^{(k)} \right\}^2 - \frac{1}{N} \sum_{k=1}^{N} \left\{ \mathcal{N}\left(h^{-(k)}, 4\sigma^2\right) - y^{(k)} \right\}^2 \right]$$

$$= c_i^{(l)} - \frac{\eta}{2\epsilon} \left[ \frac{1}{N} \sum_{k=1}^{N} \left\{ 2\sigma Z^{+(k)} + h^{+(k)} - y^{(k)} \right\}^2 - \frac{1}{N} \sum_{k=1}^{N} \left\{ 2\sigma Z^{-(k)} + h^{-(k)} - y^{(k)} \right\}^2 \right].$$

We have changed the normal variables into standard normal variables by $Z = \left( \mathcal{N}(\mu, \sigma^2) - \mu \right)/\sigma$, for ease of manipulation. Note that the random variables $Z^{+(k)}, Z^{-(k)}$ will be independent from each other, and among different training examples $k$ as well. We find the original term $\Delta c_i^{(l)} = \frac{1}{N} \sum_{k=1}^{N} \left\{ h^{+(k)} - y^{(k)} \right\}^2 - \frac{1}{N} \sum_{k=1}^{N} \left\{ h^{-(k)} - y^{(k)} \right\}^2$ from eq (27) without Gaussian noise. The terms in the square brackets of the expression above can be rearranged to be equivalent to this $\Delta c_i^{(l)}$ plus the following random variables:

$$4\sigma \frac{1}{N} \sum_{k=1}^{N} \left\{ \sigma\left(\chi_1^2 - \chi_1^2\right) + \left(h^{+(k)} - y^{(k)}\right) Z^{+(k)} - \left(h^{-(k)} - y^{(k)}\right) Z^{-(k)} \right\}$$

$$= 4\sigma^2 \frac{1}{N} \left(\chi_N^2 - \chi_N^2\right) + 4\sigma \frac{1}{N} \sum_{k=1}^{N} \left\{ \mathcal{N}\left(0, \left(h^{+(k)} - y^{(k)}\right)^2\right) - \mathcal{N}\left(0, \left(h^{-(k)} - y^{(k)}\right)^2\right) \right\}$$

$$= 4\sigma^2 \frac{1}{N} \left(\chi_N^2 - \chi_N^2\right) + 4\sigma \frac{1}{N} \sum_{k=1}^{N} \left\{ \mathcal{N}\left(0, \left(h^{+(k)} - y^{(k)}\right)^2 + \left(h^{-(k)} - y^{(k)}\right)^2\right) \right\}.$$

With $\chi_N^2$ being the chi-squared distrbution with N degrees of freedom, and $Z^2 \sim \chi_1^2$, and $\sum_{k=1}^{N} \chi_1^2 = \chi_N^2$. The two $\chi_N^2$ are each from the original term containing $h^{+(k)}$ and $h^{-(k)}$ respectively. In the large limit, $\lim_{N \to \infty} \chi_N^2 \approx \mathcal{N}(N, 2N)$. As such, the above expression becomes:

$$4\sigma^2 \frac{1}{N} \left(\mathcal{N}(N, 2N) - \mathcal{N}(N, 2N)\right) + 4\sigma \frac{1}{N} \sum_{k=1}^{N} \left\{ \mathcal{N}\left(0, \left(h^{+(k)} - y^{(k)}\right)^2 + \left(h^{-(k)} - y^{(k)}\right)^2\right) \right\}$$

$$= 4\sigma^2 \frac{1}{N} \mathcal{N}(0, 4N) + 4\sigma \frac{1}{N} \sum_{k=1}^{N} \left\{ \mathcal{N}\left(0, \left(h^{+(k)} - y^{(k)}\right)^2 + \left(h^{-(k)} - y^{(k)}\right)^2\right) \right\}$$

$$= \mathcal{N}\left(0, \frac{64\sigma^2}{N}\right) + \mathcal{N}\left(0, 16\sigma^2 \frac{1}{N^2} \sum_{k=1}^{N} \left\{ \left(h^{+(k)} - y^{(k)}\right)^2 + \left(h^{-(k)} - y^{(k)}\right)^2 \right\}\right)$$

$$= 4\sigma \mathcal{N}\left(0, \frac{4\sigma^2}{N} + \frac{1}{N^2} \sum_{k=1}^{N} \left\{ \left(h^{+(k)} - y^{(k)}\right)^2 + \left(h^{-(k)} - y^{(k)}\right)^2 \right\}\right).$$

For the quantum dataset, $N = 51$. To illustrate that the approximation $\lim_{N \to \infty} \chi_N^2 \approx \mathcal{N}(N, 2N)$ is valid for this value, we simulate 10000 values of $\frac{1}{N} \sum_{k=1}^{N} \left(\chi_1^2 - \chi_1^2\right)$ and plot the histogram of these values against the expected pdf $\mathcal{N}(0, 4/N)$.

Thus, when Gaussian noise is introduced, the value by which a parameter $c_i^{(l)}$ shifts in an epoch will be distributed about its noiseless value $\Delta c_i^{(l)}$ with variance of $\left(\frac{2\sigma\eta}{\epsilon}\right)^2 \left[\frac{4\sigma^2}{N} + \frac{1}{N^2} \sum_{k=1}^{N} \left\{ \left(h^{+(k)} - y^{(k)}\right)^2 + \left(h^{-(k)} - y^{(k)}\right)^2 \right\}\right]$. I.e. eq. (27) will become the below expression

$$c_i^{(l)} \leftarrow c_i^{(l)} - \mathcal{N}\left( \Delta c_i^{(l)}, \left(4\sigma \frac{\eta}{2\epsilon}\right)^2 \left[\frac{4\sigma^2}{N} + \frac{1}{N^2} \sum_{k=1}^{N} \left\{ \left(h^{+(k)} - y^{(k)}\right)^2 + \left(h^{-(k)} - y^{(k)}\right)^2 \right\}\right] \right). \quad \text{(A.1)}$$

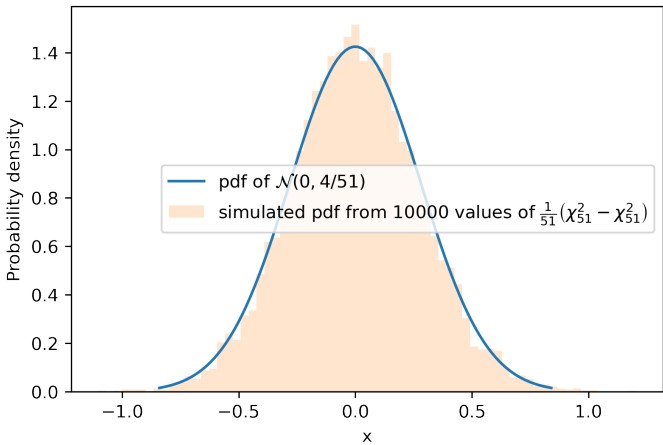

Figure 23: Estimated pdf from 10000 monte-carlo simulations of $\frac{1}{N}\sum_{k=1}^{N}\sigma\left(\chi_1^2-\chi_1^2\right)$ plotted as a histogram, compared to the approximate pdf $\mathcal{N}(0,4/N)$.

The predicted values and the target values are both limited to be within $-1 \geq h^{+(k)}, h^{-(k)}, y^{(k)} \geq 1$. The variance therefore has an upper bound of $\left(\frac{2\sigma\eta}{\epsilon}\right)^2\left[\frac{4\sigma^2+8}{N}\right]$.

## B Repeated tests of DPC for quantum dataset

We plot the training-loss graph for the noiseless case and the $p = 0.05$ case repeated over 10 different sets of initial parameters (generated using random seed = 0-9) in figure (24), and tabulate the final loss over the training and test sets in table (8) below.

The loss on both training and test sets are higher in the $p = 0.05$ case compared to the noiseless case, and the lower loss we observed in section (5.2) for the singular $p = 0.05$ case is likely due to the minute differences in how parameters are updated during the gradient descent process from the action of the DPC, as elaborated in section (5.2).

Note that the loss increases during some runs of the noiseless case around epoch 42. This is due to some parameters overshooting the minima. We see similar increases at epoch 45 for the $p = 0.05$ case. This can be explained by the reduction in the value by which parameters are shifted during gradient descent, as mentioned in section (5.2) that these are scaled by $(1-p)$ on average in the presence of DPC. The parameters would take slightly longer to reach the minima, hence the overshoot occurs at a later epoch.

We believe the initial singular instance of $p = 0.05$ achieving a lower loss than the noiseless case can be explained by this reason as well. Some parameter in the noiseless case might have overshot the minima slightly, but shifts in other parameters still reduced the loss compared to the previous epoch. Since most parameter updates are scaled by $(1-p)$ on average, this parameter might not have overshot yet by epoch 50, which leads to the lower loss.

Table 8: The loss (true MSE, without DPC applied) on the training and test datasets for the trained model after 50 training epochs.

| $p$ | Loss (Training set) | Loss (Test set) |
|------|---------------------|-----------------|
| 0 | $0.02626 \pm 0.00090$ | $0.0199 \pm 0.0013$ |
| 0.05 | $0.0273 \pm 0.0014$ | $0.0201 \pm 0.0014$ |

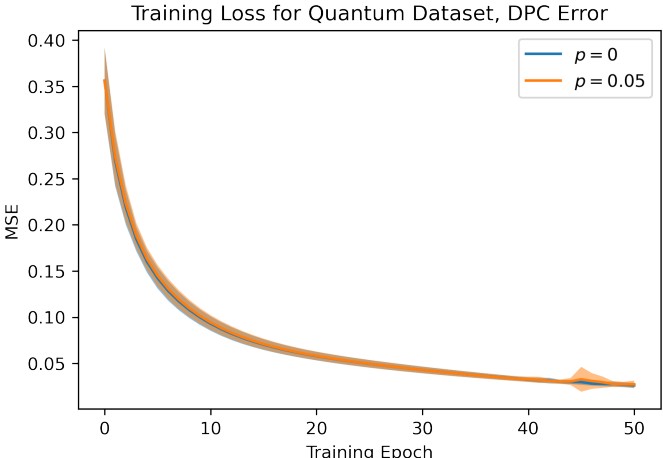

Figure 24: The loss (true MSE over the training set, without any DPC applied) between the noiseless and $p = 0.05$ case, calculated at the end of every training epoch during the gradient descent process. The solid lines indicate the mean loss at each epoch over the 10 repeated simulations, while the shaded regions indicate the standard deviation.

## C  Effect of DPC noise on parameter update with MSE loss

With DPC noise of magnitude $p$, a parameter update in the gradient process process from eq. (24) would be as:

$$c_i^{(l)} \leftarrow c_i^{(l)} - \eta \frac{\partial L((1-p)h[|\psi\rangle_{\text{in}}^{(1,2,...N)};\mathbf{c}], y^{(1,2,...,N)})}{\partial c_i^{(l)}}. \tag{C.1}$$

Considering we estimate the partial derivative in the expression above via the finite difference method as stipulated in eq. (23), and a mean squared error loss function as defined in eq. (26), the above expression can be written explicitly:

$$c_i^{(l)} \leftarrow c_i^{(l)} - \frac{\eta}{2\epsilon}\left[L\left((1-p)h^{+(1,2,...,N)}, y^{(1,2,...,N)}\right) - L\left((1-p)h^{-(1,2,...,N)}, y^{(1,2,...,N)}\right)\right]$$

$$= c_i^{(l)} - \frac{\eta}{2\epsilon}\left[\frac{1}{N}\sum_{k=1}^{N}\left\{(1-p)h^{+(k)} - y^{(k)}\right\}^2 - \frac{1}{N}\sum_{k=1}^{N}\left\{(1-p)h^{-(k)} - y^{(k)}\right\}^2\right]. \tag{C.2}$$

The notation is identical to the case in eq. (27). From which we can compare the value of the gradient in the DPC case $\left(\frac{\partial L}{\partial c_i^{(l)}}\right)_p$, in eq. (C.2), as a ratio to what the gradient would be without DPC $\left(\frac{\partial L}{\partial c_i^{(l)}}\right)_{p=0}$, in eq. (27)

$$\frac{\left(\frac{\partial L}{\partial c_i^{(l)}}\right)_p}{\left(\frac{\partial L}{\partial c_i^{(l)}}\right)_{p=0}} = \frac{\sum_{k=1}^{N}\left\{(1-p)h^{+(k)} - y^{(k)}\right\}^2 - \sum_{k=1}^{N}\left\{(1-p)h^{-(k)} - y^{(k)}\right\}^2}{\sum_{k=1}^{N}\left\{h^{+(k)} - y^{(k)}\right\}^2 - \sum_{k=1}^{N}\left\{h^{-(k)} - y^{(k)}\right\}^2}$$

$$= 1 - p\frac{\sum_{k=1}^{N}\left\{2y^{(k)}(h^{+(k)} - h^{-(k)}) + (p-2)\left(\left(h^{+(k)}\right)^2 - \left(h^{-(k)}\right)^2\right)\right\}}{\sum_{k=1}^{N}\left\{2y^{(k)}(h^{+(k)} - h^{-(k)}) - \left(\left(h^{+(k)}\right)^2 - \left(h^{-(k)}\right)^2\right)\right\}}$$

$$= 1 - p \left[ 1 - \frac{(1-p) \sum_{k=1}^{N} \left\{ \left(h^{+(k)}\right)^2 - \left(h^{-(k)}\right)^2 \right\}}{\sum_{k=1}^{N} \left\{ 2 y^{(k)} (h^{+(k)} - h^{-(k)}) - \left( \left(h^{+(k)}\right)^2 - \left(h^{-(k)}\right)^2 \right) \right\}} \right]$$

$$= 1 - p \left[ 1 + \frac{(1-p) \sum_{k=1}^{N} \left\{ \left(h^{+(k)} - h^{-(k)}\right) \left(h^{+(k)} + h^{-(k)}\right) \right\}}{\sum_{k=1}^{N} \left\{ \left(h^{+(k)} - h^{-(k)}\right) \left(h^{+(k)} + h^{-(k)} - 2 y^{(k)}\right) \right\}} \right].$$

## D  Effect of gaussian noise on parameter update with cross-entropy/logloss

We begin with the parameter update process from eq. (29), which would become the following with the introduction of gaussian noise:

$$c_i^{(l)} \leftarrow c_i^{(l)} - \frac{\eta}{2\epsilon} \left[ \frac{1}{N} \sum_{k=1}^{N} \left\{ y^{(k)} \log_2 \left( \frac{\mathcal{N}\left(h^{-(k)}, \sigma^2\right)}{\mathcal{N}\left(h^{+(k)}, \sigma^2\right)} \right) + (1 - y^{(k)}) \log_2 \left( \frac{1 - \mathcal{N}\left(h^{-(k)}, \sigma^2\right)}{1 - \mathcal{N}\left(h^{+(k)}, \sigma^2\right)} \right) \right\} \right].$$
(D.1)

There is no factor of 4 here, as compared to the quantum dataset case, for the variance in the predicted values $h^{+(k)}, h^{-(k)}$, as we use the probability of measuring the state $|+\rangle$ rather than the expectation value of measuring in the $x$-basis as we did in the quantum dataset case.

We begin with the term in the estimated gradient of eq. (D.1), leaving out the factor $\frac{\eta}{2\epsilon}$ for simplicity,

$$\frac{1}{N} \sum_{k=1}^{N} \left\{ y^{(k)} \log_2 \left( \frac{\mathcal{N}\left(h^{-(k)}, \sigma^2\right)}{\mathcal{N}\left(h^{+(k)}, \sigma^2\right)} \right) + (1 - y^{(k)}) \log_2 \left( \frac{1 - \mathcal{N}\left(h^{-(k)}, \sigma^2\right)}{1 - \mathcal{N}\left(h^{+(k)}, \sigma^2\right)} \right) \right\}$$

$$= \frac{1}{N} \sum_{k=1}^{N} \left\{ y^{(k)} \log_2 \left( \frac{h^{-(k)} + \mathcal{N}\left(0, \sigma^2\right)}{h^{+(k)} + \mathcal{N}\left(0, \sigma^2\right)} \right) + (1 - y^{(k)}) \log_2 \left( \frac{1 - h^{-(k)} - \mathcal{N}\left(0, \sigma^2\right)}{1 - h^{+(k)} - \mathcal{N}\left(0, \sigma^2\right)} \right) \right\}$$

$$= \frac{1}{N} \sum_{k=1}^{N} \left\{ y^{(k)} \log_2 \left( \frac{h^{-(k)}}{h^{+(k)}} \frac{1 + \mathcal{N}\left(0, \frac{\sigma^2}{\left(h^{-(k)}\right)^2}\right)}{1 + \mathcal{N}\left(0, \frac{\sigma^2}{\left(h^{+(k)}\right)^2}\right)} \right) + (1 - y^{(k)}) \log_2 \left( \frac{1 - h^{-(k)}}{1 - h^{+(k)}} \frac{1 + \mathcal{N}\left(0, \frac{\sigma^2}{\left(1 - h^{-(k)}\right)^2}\right)}{1 + \mathcal{N}\left(0, \frac{\sigma^2}{\left(1 - h^{+(k)}\right)^2}\right)} \right) \right\}.$$

From the second to third line, we have switched the signs of the normal variable in the second term, since the pdf of a normal distribution with mean 0 is an even function. For the small values of $\sigma$ that we tested, $\sigma << h^{+(k)}, h^{-(k)}$, since typical values of $h$ are distributed around 0.5. We can apply the approximation suggested in [41], where $\ln(1 + X) \approx X - \frac{1}{2}X^2 + O(X^3)$ for a random normal variable $X$ distributed very close to 0, i.e. with variance $\sigma^2 << 1$.

$$= \frac{1}{N} \sum_{k=1}^{N} \left\{ y^{(k)} \left[ \log_2 \left( \frac{h^{-(k)}}{h^{+(k)}} \right) + \frac{1}{\ln 2} \ln \left( 1 + \mathcal{N}\left(0, \frac{\sigma^2}{\left(h^{-(k)}\right)^2}\right) \right) - \frac{1}{\ln 2} \ln \left( 1 + \mathcal{N}\left(0, \frac{\sigma^2}{\left(h^{+(k)}\right)^2}\right) \right) \right] \right.$$

$$+ (1 - y^{(k)}) \left[ \log_2 \left( \frac{1 - h^{-(k)}}{1 - h^{+(k)}} \right) + \frac{1}{\ln 2} \ln \left( 1 + \mathcal{N}\left(0, \frac{\sigma^2}{\left(1 - h^{-(k)}\right)^2}\right) \right) - \frac{1}{\ln 2} \ln \left( 1 + \mathcal{N}\left(0, \frac{\sigma^2}{\left(1 - h^{+(k)}\right)^2}\right) \right) \right] \right\}$$

$$\approx \frac{1}{N} \sum_{k=1}^{N} \left\{ y^{(k)} \left[ \log_2 \left( \frac{h^{-(k)}}{h^{+(k)}} \right) + \frac{1}{\ln 2} \mathcal{N}\left(0, \frac{\sigma^2}{\left(h^{-(k)}\right)^2}\right) - \frac{1}{\ln 2} \mathcal{N}\left(0, \frac{\sigma^2}{\left(h^{+(k)}\right)^2}\right) \right] \right.$$

$$+ (1 - y^{(k)}) \left[ \log_2 \left( \frac{1 - h^{-(k)}}{1 - h^{+(k)}} \right) + \frac{1}{\ln 2} \mathcal{N}\left(0, \frac{\sigma^2}{\left(1 - h^{-(k)}\right)^2}\right) - \frac{1}{\ln 2} \mathcal{N}\left(0, \frac{\sigma^2}{\left(1 - h^{+(k)}\right)^2}\right) \right] \right\}$$

$$= \frac{1}{N} \sum_{k=1}^{N} \left\{ y^{(k)} \left[ \log_2 \left( \frac{h^{-(k)}}{h^{+(k)}} \right) + \frac{1}{\ln 2} \mathcal{N} \left( 0, \frac{\sigma^2}{\left(h^{-(k)}\right)^2} + \frac{\sigma^2}{\left(h^{+(k)}\right)^2} \right) \right] \right.$$
$$\left. + (1 - y^{(k)}) \left[ \log_2 \left( \frac{1 - h^{-(k)}}{1 - h^{+(k)}} \right) + \frac{1}{\ln 2} \mathcal{N} \left( 0, \frac{\sigma^2}{\left(1 - h^{-(k)}\right)^2} + \frac{\sigma^2}{\left(1 - h^{+(k)}\right)^2} \right) \right] \right\}.$$

We can recover the original, noiseless term

$$\Delta c_i^{(l)} = \frac{1}{N} \sum_{k=1}^{N} \left\{ y^{(k)} \log_2 \left( \frac{h^{-(k)}}{h^{+(k)}} \right) + (1 - y^{(k)}) \log_2 \left( \frac{1 - h^{-(k)}}{1 - h^{+(k)}} \right) \right\},$$

alike the MSE scenario in appendix (A). We isolate the random normal variables that are added on top of this term:

$$\frac{1}{N} \sum_{k=1}^{N} \left\{ y^{(k)} \left[ \frac{1}{\ln 2} \mathcal{N} \left( 0, \frac{\sigma^2}{\left(h^{-(k)}\right)^2} + \frac{\sigma^2}{\left(h^{+(k)}\right)^2} \right) \right] + (1 - y^{(k)}) \left[ \frac{1}{\ln 2} \mathcal{N} \left( 0, \frac{\sigma^2}{\left(1 - h^{-(k)}\right)^2} + \frac{\sigma^2}{\left(1 - h^{+(k)}\right)^2} \right) \right] \right\}$$
$$= \frac{\sigma}{N \ln 2} \mathcal{N} \left( 0, \sum_{k=1}^{N} \left\{ (y^{(k)})^2 \left[ \frac{1}{\left(h^{-(k)}\right)^2} + \frac{1}{\left(h^{+(k)}\right)^2} \right] + \left(1 - y^{(k)}\right)^2 \left[ \frac{1}{\left(1 - h^{-(k)}\right)^2} + \frac{1}{\left(1 - h^{+(k)}\right)^2} \right] \right\} \right).$$

Note that the factor $\frac{\eta}{2\epsilon}$ is still left out in the above term. The variance about $\Delta c_i^{(l)}$ would therefore be:

$$\left( \frac{\eta}{2\epsilon} \frac{\sigma}{N \ln 2} \right)^2 \sum_{k=1}^{N} \left\{ (y^{(k)})^2 \left[ \frac{1}{\left(h^{-(k)}\right)^2} + \frac{1}{\left(h^{+(k)}\right)^2} \right] + \left(1 - y^{(k)}\right)^2 \left[ \frac{1}{\left(1 - h^{-(k)}\right)^2} + \frac{1}{\left(1 - h^{+(k)}\right)^2} \right] \right\}.$$

Since the predicted values are bounded by $0 \leq h^{+(k)}, h^{-(k)} \geq 1$, the above expression has a lower bound of $\left( \frac{\eta\sigma}{2(\ln 2)\epsilon} \right)^2 \frac{1}{N}$.

## E  Effect of DPC noise on parameter update with cross-entropy/logloss

The output of the QCNN for the classical dataset is given as the probability of measuring the state $|+\rangle$ rather than the expectation value of a measurement in the $x$-basis, as explained in section (11.2). Considering the effect of the DPC in eq. (12), this has the effect of scaling the predicted value towards $1/2$ proportional to $p$:

$$h[|\psi\rangle_{in}; \mathbf{c}] \rightarrow (1 - p)h[|\psi\rangle_{in}; \mathbf{c}] + \frac{p}{2}. \tag{E.1}$$

This will affect the parameter update term in the gradient descent process, from eq. (29):

$$c_i^{(l)} \leftarrow c_i^{(l)} - \frac{\eta}{2\epsilon} \left[ \frac{1}{N} \sum_{k=1}^{N} \left\{ y^{(k)} \log_2 \left( \frac{(1-p)h^{-(k)} + \frac{p}{2}}{(1-p)h^{+(k)} + \frac{p}{2}} \right) + (1 - y^{(k)}) \log_2 \left( \frac{1 - (1-p)h^{-(k)} - \frac{p}{2}}{1 - (1-p)h^{+(k)} - \frac{p}{2}} \right) \right\} \right]. \tag{E.2}$$

The gradient term within the square brackets can be manipulated into:

$$\frac{1}{N} \sum_{k=1}^{N} \left\{ y^{(k)} \log_2 \left( \frac{h^{-(k)} + \frac{p}{2(1-p)}}{h^{+(k)} + \frac{p}{2(1-p)}} \right) + (1 - y^{(k)}) \log_2 \left( \frac{1 - h^{-(k)} + \frac{p}{2(1-p)}}{1 - h^{+(k)} + \frac{p}{2(1-p)}} \right) \right\}. \tag{E.3}$$

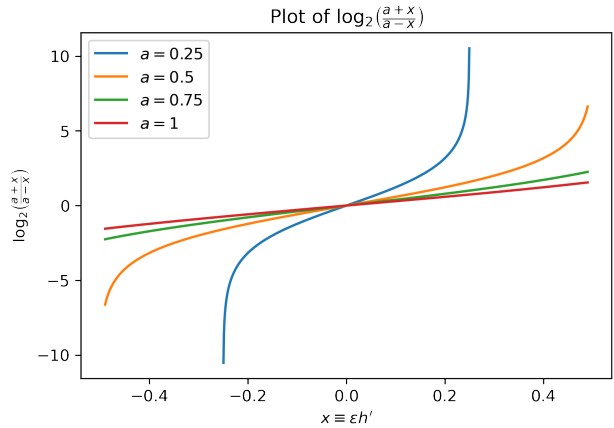

Figure 25: The logarithm of the fraction $\frac{a+x}{a-x}$ at various values of $a$ that are typical in eq. (29) and eq. (E.3). The domain of $x$ of each curve is limited to the condition that $|x| < a$.

Keep in mind that $h^{+(k)}, h^{-(k)}$ are terms meant to approximate the gradient of the predicted value $h \equiv h[|\psi\rangle_{in}; \mathbf{c}]$ through the finite difference method; $h^{+(k)} \approx h + \epsilon h' + O(\epsilon^2)$, $h^{-(k)} \approx h - \epsilon h' + O(\epsilon^2)$, with $h'$ being the first order partial derivative of $h$ with respect to the parameter $c_i^{(l)}$. Given this, the logarithmic terms from eq. (29) can also be expressed as:

$$
y^{(k)} \log_2\left(\frac{h^{-(k)}}{h^{+(k)}}\right) + (1 - y^{(k)}) \log_2\left(\frac{1 - h^{-(k)}}{1 - h^{+(k)}}\right)
$$
$$
\approx y^{(k)} \log_2\left(\frac{h - \epsilon h' + O(\epsilon^2)}{h + \epsilon h' + O(\epsilon^2)}\right) + (1 - y^{(k)}) \log_2\left(\frac{1 - h + \epsilon h' - O(\epsilon^2)}{1 - h - \epsilon h' - O(\epsilon^2)}\right). \tag{E.4}
$$

The terms inside the logarithms can therefore be thought of as some $\frac{a+x}{a-x}$, with $x \equiv \epsilon h'$ the term that gives the difference between $h^{+(k)}, h^{-(k)}$, and $a$ as some constant that is common between the numerator and denominator - $h$ in the first logarithm and $1 - h$ in the second logarithm of eq (E.3). For clarity, we only regarded terms up to first order in $\epsilon$ here. The term $x$ can very well also represent the terms with odd powers of $\epsilon$: $\epsilon h', \epsilon^3 h''', ...$, and $a$ can include the even powers of $\epsilon$ as well. Furthermore, $0 < a < 1, |x| < a$ to keep the condition that all predicted values $h^{+(k)}, h^{-(k)}$, and $h$ lie between zero and one.

From eq. (E.3), the action of the DPC would be akin to shifting the terms in the logarithms to $\frac{a + x + \frac{p}{2(1-p)}}{a - x + \frac{p}{2(1-p)}}$, from some original $\frac{a+x}{a-x}$.

As we observe in Figure (25) above, at every value of $x$, the value of $\log_2\left(\frac{a+x}{a-x}\right)$ decreases as $a$ increases. The terms in the DPC case in eq. (E.3) will always have a lower absolute value compared to their equivalents in the noiseless case, i.e.:

$$
\left| \log_2\left(\frac{h^{-(k)} + \frac{p}{2(1-p)}}{h^{+(k)} + \frac{p}{2(1-p)}}\right) \right| < \left| \log_2\left(\frac{h^{-(k)}}{h^{+(k)}}\right) \right|,
$$
$$
\left| \log_2\left(\frac{1 - h^{-(k)} + \frac{p}{2(1-p)}}{1 - h^{+(k)} + \frac{p}{2(1-p)}}\right) \right| < \left| \log_2\left(\frac{1 - h^{-(k)}}{1 - h^{+(k)}}\right) \right|.
$$

Hence, the values by which the parameters will be updated with DPC in eq. (E.3) will be lower on average, compared to a noiseless ($p = 0$) but otherwise exactly equivalent case, at the same values of parameters, inputs, etc. The difference will be larger if $p$ is larger as well.

## F Scaled value of parameter with DPC with learning rate

The learning rate $\eta$ begins at a value of 1 and is increased by 1.05 every epoch that the loss does not increase. If we assume the gradient remains at a constant $\Delta c_i^{(l)}$ throughout the process, the cumulative distance after 12 epochs in the noiseless case will be:

$$\sum_{j=1}^{12}(1.05)^{j-1}\Delta c_i^{(l)} = 15.9\Delta c_i^{(l)}.$$

If we also assume that the gradient for the same parameter is also constant for the $p = 0.5$ case, over 31 epochs, the cumulative distance when this gradient is scaled by some constant $k_p$ will be:

$$\sum_{j=1}^{31}(1.05)^{j-1}k_p\Delta c_i^{(l)} = 70.8k_p\Delta c_i^{(l)}.$$

For the two distances above to be equal, we see that $k_p = 0.225$. In reality, the gradient might decrease as it approaches the minima in the later epochs, and this might offset the greater value of the learning rate in the later epochs - the original estimate for $1/3$ might be slightly more realistic than this.

## G Cumulative distance of parameter over 31 epochs with DPC

The learning rate $\eta$ begins at a value of 1 and is increased by 1.05 every epoch that the loss does not increase. For the $p = 0$ case, the loss increases at epoch 12 and 18. If we also assume gradients are constant at $\Delta c_i^{(l)}$ over all training epochs for all parameters, in the noiseless case, the cumlative distance covered by $c_i^{(l)}$ over the 31 epochs will be:

$$\sum_{j=1}^{12}(1.05)^{j-1}\Delta c_i^{(l)} + \sum_{j=12}^{18}\frac{1}{2}(1.05)^{j-2}\Delta c_i^{(l)} + \sum_{j=18}^{31}\frac{1}{2}(1.05)^{j-3}\Delta c_i^{(l)} = 31.4\Delta c_i^{(l)}.$$

In the $p = 0.5$ case, if we take a gradient that is scaled by the average factor of 0.53,

$$\sum_{j=1}^{31}(1.05)^{j-1}0.53\Delta c_i^{(l)} = 37.524k_p\Delta c_i^{(l)}.$$

Which is larger than that in the $p = 0$ case! The parameter in the $p = 0.5$ case will shift a greater cumulative distance than the $p = 0$ case. However, oftentimes, the gradient might decrease over training as it gets closer to the minima, which might once again reduce the effect of the increased learning rate in the later epochs, and the distance might be closer to the $p = 0$ than shown.

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
