# Peer review of "Quantifying the Effects of Noise in a Quantum Convolutional Neural Network"

_SciPost Physics, doi:SciPost Phys. Core 8, 093 (2025)_

## Round 1 · Referee Report · Anonymous (Referee 1) · 2024-8-19

Strengths

1- Analytical calculation of the effect of Gaussian noise on the distribution of gradients observed during the QCNN training process. 2- Analytical calculation of the effect of the depolarizing channel on the gradient of the loss function for the quantum dataset.

Weaknesses

1- In its current form, the manuscript does not provide much insight into the question whether noise can enhance the training performance of QCNNs.

Report

The authors study the effect of noise on the training procedure of quantum convolutional neural nets (QCNNs). In particular, they investigate three types of noise, Gaussian noise on the probability of measuring the output qubit in a certain state, a depolarizing channel applied before the final measurement, and coherent single-qubit state preparation errors designed to affect the training process at each step in a maximally adverse way. Using a QCNN with up to nine qubits, they examine the training process under the aforementioned types of noise for both a classical data set (subset of the MNIST data set) and a quantum data set using classical state vector simulations of the corresponding quantum circuits. Moreover, an analytical expression how Gaussian noise affects the distribution of gradients observed during the training process is derived. In general, for both data sets no significant benefit for the training process for any of the three types of noises studied is observed.

Although it is definitely an interesting question whether quantum noise can be beneficial in the training process of QCNNs and more general quantum learning models, the results in the current manuscript do not shed much light on this question in my opinion. Below are some detailed comments and questions regarding the manuscript.

1- I think the presentation should be improved in various aspects.

Some equation labels do not seem to be properly defined, and references to the corresponding equations direct to a section number or show up as "??". Similarly, there are various references to sections/appendices that direct to (sub-)sections that not seem to exist, and what is referred to appendix is instead titled “Methods & Implementation”. Below is a list of these issues. • page 4, right column, first sentence (eq. (III)) • page 6, right column at the end of the first paragraph (Appendix (A)) • page 7, right column, paragraph below Table II (Appendix (B)) • page 7, right column, below Eq. (14) (Appendix (C)) • page 20, Sec. XI, in the subsection "Training and Test Datasets" (section (??)) • page 21, Sec. XI, in the subsection “Loss Function”, towards the end of the page, third bullet point (eq. (??)) • page 22, at the end of Sec. XII (Appendix (A)); • page 22, right at the beginning of the subsection "Quantum Depolarizing Channel" (eq. (III)) Various quantities in the main text are not properly defined, e.g. $h^{\pm(k)}$, $\lambda$. While these quantities show up in “Methods & Implementation”, it would help to at least mention in the main text what they are. The authors state that towards the end of Sec. V that “… we observe a lightly faster decrease in loss in the first few epochs at small values of $\bar{\delta}$ …” and refer to Fig. 10. However, this is not visible in the current version of Fig. 10, I would suggest an inset for the first few epochs to show the effect clearly. The labels in Fig. 19 are too small to be properly visible in a printed copy. I suggest stacking the two panels below each other. Moreover panel (a) does have not have any axes labeling.

2- The authors restrict themselves to a QCNNs with up to 9 qubits and perform state vector simulations, corresponding to a quantum computer allowing for an infinite number of measurements. Do they expect their results would change for a larger number of qubits? What prevents a classical simulation with up to $\mathcal{O}(20)$ qubits which should still be tractable on a classical computer and would allow for studying the behavior with system size? What would happen in a realistic scenario with a finite number of measurements?

3- The main problem, in my opinion, is there is no clear finding regarding the performance enhancement of QCNNs through the introduction of noise. As stated in the conclusion of the manuscript, “there is insufficient evidence to confidently conclude that noise may be beneficial ….”. In fact, in all cases studied for the quantum dataset, there is clearly no improvement. The improvements observed for the classical dataset seem to be within statistical fluctuations. Given that the authors studied an idealized scenario performing a state vector simulation with very specific types of noise, one can neither draw conclusions whether noise would be beneficial provided one has sufficient control over it, nor how training a QCNN on a real quantum device would perform.

4- Minor typos: - In the introduction on page 1, left column towards the end of the page the word "quantum" in "Optical quantum qubits... " seems to be superfluous. - Caption of Table IV: epochs should have no capital letter at the beginning - Page 13, left column, last paragraph: “ …, due likely to the fact …” should be “ …, likely due to the fact …”

Unfortunately, I think the present form of the manuscript does not meet any of the four acceptance criteria, and I cannot recommend it for publication in SciPost Physics.

Requested changes

see report

Recommendation

Reject

---

## Round 1 · Referee Report · Anonymous (Referee 2) · 2024-11-7

Strengths

1 - Analytical calculation of the effects of Gaussian noise and a quantum depolarizing channel on the gradients of the neural network
2 - In-depth analysis and discussion of each result

Weaknesses

1 - The result the authors obtain with their models is "negative", i.e., the noises they consider are not beneficial for the QCNN they consider and the problems they tackle.

Report

The authors analyse the effect of three type of noises, namely Gaussian noise, Quantum Depolarizing Channel and perturbation of inputs states, on the performance of a QCNN. They use a 9 qubits QCNN, which they simulate on a classical computer. They analyse one problem with a quantum dataset, the Quantum Phase Recognition task, and one problem with classical dataset, the MNIST dataset for handwritten digit recognition.

While the results are "negative" in the sense that they do not find any improvement on the performance of the QCNN when noise is added (actually the performance degrades most of the times), the analysis is interesting and in-depth. I find very interesting also the deep analysis of the effect of the noises on the training procedure, with which they are able to find strategies to avoid the detrimental effects of noise.

In my opinion this manuscript does not met the criteria of SciPost Physics, but it could meet the criteria of SciPost Physics Core if the following comments are properly addressed.

Requested changes

1 - In general I find the manuscript quite hard to read, I think the authors should improve the presentation.
2 - When they describe the Quantum Dataset, they have to describe how the input to the QCNN is encoded, how they do for the MNIST dataset. Furthermore they should also indicate the output of the QCNN for clarity and completeness.
3 - Pag. 4, first column, unnumbered equation between (7) and (8): the 2 at the denominator is correct only if the system is a qubit. Here they are introducing the DPC for a general quantum system and 2 should be substituted with the dimension of the Hilbert space.
4 - Page 8, beginning of the first column: The authors state that the coefficient of $p$ is distributed around a value slightly above 1. However, this coefficient should actually be distributed around a value slightly below 1, so that the entire expression centers around a value slightly above $1−p$, as depicted in Fig. 9.
5 - In the conclusion, the authors state, "There is insufficient evidence to confidently conclude that noise may be beneficial in the operation of the QCNN we had tested" and "Compared to the MNIST dataset, we were unable to find any benefit in applying quantum noise to our quantum dataset whatsoever," which implies that noise may be beneficial for the MNIST dataset. In my opinion, this interpretation stretches the results: the findings clearly show that, for the specific noise models, QCNN architecture, and tasks examined (Quantum Phase Recognition and MNIST), noise generally degrades performance rather than enhances it. The minor improvements observed in a few isolated instances do not exceed statistical fluctuations, and the claimed benefits (such as those for the Depolarizing Channel applied to MNIST, see Fig.17 and the associated discussion) remain speculative. I recommend that the authors revise these claims for clarity and accuracy.
6 - The acknowledgements are the acknowledgements of a thesis and also an author seems to be thanked.

Recommendation

Accept in alternative Journal (see Report)

---

## Round 2 · Referee Report · Anonymous (Referee 2) · 2025-10-8

The referee discloses that the following generative AI tools have been used in the preparation of this report:
report written by myself, then polished with ChatGPT 4o, then fixed by me again.
Report
Recommendation
Accept in alternative Journal (see Report)

---

## Round 2 · Referee Report · Anonymous (Referee 3) · 2025-11-12

The referee discloses that the following generative AI tools have been used in the preparation of this report:
I used generative AI tools to improve the writing of the report
Report
The thorough examination of both a quantum phase recognition task and a classical image classification task provides an extensive viewpoint. I consider the data and analysis presented in the manuscript to be strong and a true addition to the field
However, I also see fundamental limitations that make me think that the manuscript is better suited for a journal like SciPost Physics Core. Indeed, the core finding is that quantum noise (Gaussian, Depolarizing Channel) is generally detrimental to training. While this should not be taken for granted (negative results are results and can be very valuable), I do not foresee an immediate impact of such exploratory work. Also, in the same direction, while the study is focused on how to quantify the problem of noise effect, the solutions offered are limited.
Recommendation
Accept in alternative Journal (see Report)

---

## Round 2 · List of Changes

-
Change in topic from finding advantages in introducing noise to the QCNN, to quantifying the effects of noise in the QCNN, as well as some proposals for mitigating their impacts.
-
The "Introduction" and "Datasets" sections have been expanded to include key details such as the inputs to the QCNN and how outputs are retrieved, which were previously only described in the "Methodology & Implementation" section. This is intended to make the main text more self-contained and easier to follow.
-
Modified Figures 10 and 18 for better clarity.
-
Reframed language throughout the Results, Discussion, and Conclusion sections to clarify that the focus is on quantifying the effects of noise on QCNN training and performance, rather than investigating potential benefits.
-
Removed speculative statements suggesting that noise may be beneficial in specific cases (e.g., MNIST with Depolarizing Channel noise) and replaced them with descriptions of observed effects without implying performance improvement.
-
Revised the Conclusion section to emphasize quantification of noise effects and identification of conditions where noise impacts training dynamics, while avoiding interpretations beyond what is statistically supported by the results.
-
Correction of numerous typos and errors. Apologies and thank you for catching them!

---

## Editorial Decision

published